# Topology-Preserving Neural Operator Learning via Hodge Decomposition

**Dongzhe Zheng** [1]   **Tao Zhong** [1]   **Christine Allen-Blanchette** [1]

## Abstract

In this paper, we study solution operators of physical field equations on geometric meshes from a function-space perspective. We reveal that Hodge orthogonality fundamentally resolves spectral interference by isolating unlearnable topological degrees of freedom from learnable geometric dynamics, enabling an additive approximation confined to structure-preserving subspaces. Building on Hodge theory and operator splitting, we derive a principled operator-level decomposition. The result is a Hybrid Eulerian-Lagrangian architecture with an algebraic-level inductive bias we call Hodge Spectral Duality (HSD). In our framework, we use discrete differential forms to capture topology-dominated components and an orthogonal auxiliary ambient space to represent complex local dynamics. Our method achieves superior accuracy and efficiency on geometric graphs with enhanced fidelity to physical invariants. Our code is available at https://github.com/ContinuumCoder/Hodge-Spectral-Duality.

## 1. Introduction

**Problem Background**  A wide range of continuum physics models (e.g., fluid mechanics, elasticity, electromagnetic fields, and reaction-diffusion systems) can be uniformly represented as partial differential operator equations on Riemannian manifolds with a boundary (Kovachki et al., 2023). Given a finite-dimensional Riemannian manifold $(\mathcal{M}, g)$ with a boundary, physical fields on $\mathcal{M}$ are represented as differential forms of various orders: 0-forms correspond to scalar fields such as temperature or potential energy, 1-forms correspond to flux-type covector fields such as mass flow rate or current density, and 2-forms correspond to fluxes through surface elements or vorticity (Hirani, 2003; Desbrun

[1]Princeton University. Correspondence to: Dongzhe Zheng <dz5992@princeton.edu>, Christine Allen-Blanchette <ca15@princeton.edu>.

*Proceedings of the 43rd International Conference on Machine Learning*, Seoul, South Korea. PMLR 306, 2026. Copyright 2026 by the author(s).

et al., 2003; Arnold et al., 2006; 2010); a linear-algebraic primer for readers is provided in Appendix A. Within this framework, the exterior derivative $d : \Omega^k \to \Omega^{k+1}$, the codifferential $\delta : \Omega^k \to \Omega^{k-1}$, and the Hodge star operator $* : \Omega^k \to \Omega^{n-k}$ uniformly characterize gradient, divergence, curl, and Laplacian operators. Many PDEs can be written as $\mathcal{A}(u; g, \kappa, \partial\mathcal{M}, f) = 0$, where $u \in \bigoplus_k \Omega^k(\mathcal{M})$ is a multi-order differential form field, $\kappa$ is a material property tensor, $f$ is a source term, $g$ is the metric tensor, $\partial\mathcal{M}$ is the manifold boundary, and $\mathcal{A}$ is obtained by combining $d$, $\delta$, and $*$ with $\kappa$. From the perspective of operator learning, finding a numerical solution to $\mathcal{A}$ can be viewed as learning a continuous operator $\mathcal{G} : (f, u|_{\partial\mathcal{M}}, \kappa) \mapsto u$ that is reusable across meshes and geometries.

In purely Euclidean domains, neural operator methods have achieved significant successes: Fourier Neural Operators realize global convolution through low-rank spectral kernels (Li et al., 2020a), DeepONet approximates operators via dual-branch encodings (Lu et al., 2021), and PINNs embed PDE residuals directly in losses (Raissi et al., 2019). These approaches leverage regular grids and fast spectral transforms for resolution-independent operator approximation (Kovachki et al., 2023). However, many critical applications involve physical fields on Riemannian manifolds with boundaries, curvature, and non-trivial topology—including aerodynamic fields on vehicle surfaces, geophysical fields on spherical manifolds, and biological fields on organ geometries. Such quantities correspond to differential forms whose evolution is jointly constrained by cohomological structure and Riemannian metric, making them sensitive to discretization choices. Constructing neural operators on general Riemannian manifolds that are both resolution-independent and structure-preserving constitutes the core problem this work addresses.

**Research Problem and Challenges**  On a Riemannian manifold $(\mathcal{M}, g)$ with a boundary and non-trivial topology, the temporal evolution of physical fields is simultaneously constrained by two fundamentally different types of structural constraints: topological and geometric. The inherent tension between preserving global structure and resolving local dynamics constitutes the core design trade-off in the design of our method.

Topological constraints stem from Hodge theory: the ker-

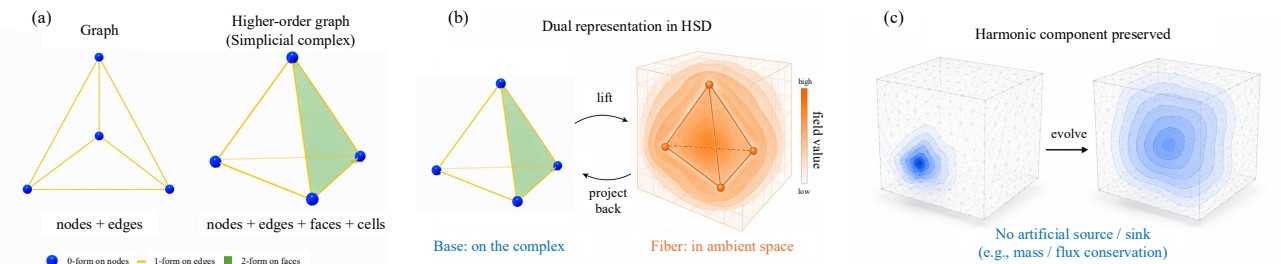

*Figure 1.* Schematic overview of the discrete representation used by HSD. (a) A simplicial complex augments a graph with higher-order cells, allowing differential-form fields to assign quantities to vertices, edges, faces, and volumes. (b) HSD represents each field through a topology-aware base on the complex and a geometry-aware fiber in ambient space. (c) The harmonic component captures the globally conserved part of a field with no artificial source or sink.

nel of the Hodge Laplacian $\Delta_k = d\delta + \delta d$ gives harmonic forms isomorphic to the $k$-th cohomology group, encoding global invariants such as circulation and net flux that must be explicitly preserved (Bhatia et al., 2012; Lim, 2020). Geometric and material constraints from the Riemannian metric $g$ and material tensor $\kappa$ govern high-frequency dynamics, diffusion anisotropy, and fine-scale structures such as boundary layers. The Hodge decomposition uniquely separates each $k$-form into gradient-type, curl-type, and harmonic components, orthogonally decoupling local differential structure from global conservation (Bhatia et al., 2012; Lim, 2020). Discrete exterior calculus and finite element exterior calculus preserve this structure exactly on simplicial complexes (Hirani, 2003; Desbrun et al., 2003; Arnold et al., 2006; 2010): vertices, edges and faces carry discrete 0-, 1-, 2-forms, and discrete operators maintain $d^2 = 0$, $\delta^2 = 0$, and Hodge decomposition. For example, Maxwell-type systems separate naturally into metric-free topological constraints such as $dF = 0$, which preserves closed-surface magnetic flux, and metric-dependent local dynamics such as $\delta F = J$, whose coefficients depend on the Hodge star and hence on geometry and material response.

Extending neural operators (Li et al., 2020a; Lu et al., 2021; Kovachki et al., 2023) to manifold settings reveals fundamental tensions. Intrinsic geometric methods based on geodesic or tangent bundle convolution (Bronstein et al., 2017) preserve manifold structure. However, they require geometry-adaptive kernels, incurring prohibitive overhead on large meshes and struggling with high-frequency patterns. Extrinsic spectral methods leverage FFT on Euclidean grids for efficient global convolution (Li et al., 2020a; Serrano et al., 2023), yet remain agnostic to cohomological and boundary topology, with limited architectural control over topological invariants. Graph-based methods rely on message passing or attention (Bronstein et al., 2017), suffering from over-smoothing or quadratic complexity, while neglecting higher-order simplicial adjacencies essential for cohomological structure (Li et al., 2018; Alon & Yahav, 2020; Wang et al., 2025). These limitations indicate that embedding the differential complex $(d, \delta, \Delta_k)$ as architectural

inductive biases while efficiently capturing high-frequency dynamics governed by metric $g$ and material tensor $\kappa$ remains open. This raises a natural question: how can operator learning on discrete meshes jointly address higher-order differential form structure across varying geometries while avoiding the efficiency–expressiveness trade-offs and topological blind spots of current approaches?

**Overview of This Work**  This paper proposes the Hodge Spectral Duality (HSD) framework for neural operator learning on oriented simplicial complexes, transforming PDE solving into structured learning on higher-order graphs with a dual-branch architecture coupled through Lie–Trotter type operator splitting (Hairer et al., 2006; Blanes et al., 2024). Our main contributions are: (1) A structure-aware neural operator framework on simplicial complexes that incorporates discrete exterior calculus as an algebraic inductive bias, ensuring physically consistent operator learning; (2) A spectral–geometric dual-branch design that separates topology-constrained global structure from geometry-driven local dynamics, enabling complementary and stable approximation of physical operators; (3) Empirical results showing improved accuracy over existing neural operator methods on complex manifold geometries, with exact preservation of cohomological invariants.

Our result reveals that Hodge orthogonality gives operator learning on manifolds an additive approximation property, enabling geometry-driven dynamics to complement topological structure while correctly completing spectral energy.

**Conflict of Interest Disclosure.**  The authors declare no conflicts of interest regarding this publication.

## 2. Related Work

**Local Methods Based on Graph and Geometric Deep Learning**  Graph-based approaches treat meshes as graphs, employing message passing or gauge equivariant convolution to approximate PDE-induced local coupling (Bronstein et al., 2017; Cohen et al., 2019; Weiler et al., 2021).

However, local aggregation mechanisms exhibit structural bottlenecks in modeling long-range dependencies: over-smoothing and over-squashing hinder networks from capturing global topological structure determined by the Hodge Laplacian kernel (Bhatia et al., 2012; Li et al., 2018; Xu et al., 2018; Oono & Suzuki, 2019; Cai & Wang, 2020; Lim, 2020; Alon & Yahav, 2020; Wang et al., 2025). Moreover, standard GNNs lack explicit encoding of differential complexes and higher-order forms, leaving algebraic identities as soft training constraints.

**Neural Operators and Spectral Methods on Manifolds**
Neural operators such as FNO and DeepONet have achieved significant progress on Euclidean domains by learning function space mappings (Raissi et al., 2019; Li et al., 2020a; Lu et al., 2021; Kovachki et al., 2023). When extending to manifolds, extrinsic embedding or background grid methods struggle to preserve intrinsic metrics and flux conservation at the discrete level (Serrano et al., 2023). Recent works attempt intrinsic operators via Laplace–Beltrami eigenbases or implicit neural fields (Serrano et al., 2023; Chen et al., 2024; Liu et al., 2025), but these primarily target scalar fields without systematically incorporating de Rham complex structure, leaving harmonic components and topological invariants implicitly entangled.

**Higher-Order Graphs, Discrete Exterior Calculus, and Topological Deep Learning** Discrete exterior calculus (DEC) and finite element exterior calculus provide rigorous frameworks for preserving algebraic and cohomological structure on simplicial complexes (Hirani, 2003; Desbrun et al., 2003; Arnold et al., 2006; 2010; Bhatia et al., 2012; Lim, 2020), including weighted Laplacian variants (Yadokoro & Bhattacharya, 2023). Topological deep learning leverages this theory for higher-order feature learning through simplicial neural networks (Papillon et al., 2023; Zia et al., 2024; Papamarkou et al., 2024; Isufi et al., 2025; Ebli et al., 2020; Chen et al., 2022; Hajij et al., 2022). However, existing works mostly focus on classification or finite-step interpolation tasks, lacking continuous operator mapping capabilities, and few explicitly separate topology-dominated from metric-dominated components.

# 3. Method: Hodge Spectral Duality Operator

This section constructs the Hodge Spectral Duality neural operator on simplicial complexes, treating signals as discrete physical fields: 0-forms on nodes (scalar potentials like temperature, pressure), 1-forms on edges (flows like velocity, current), and 2-forms on faces (fluxes like magnetic flux, vorticity).

The approach uses Hodge Decomposition under orthogonal projection via Lie-Trotter operator splitting to decouple fields into global topological and local geometric modes, with targeted neural components for each. The Hodge–de Rham decomposition $\Omega^k = \mathrm{im}(d_{k-1}) \oplus \mathrm{im}(\delta_{k+1}) \oplus \ker(\Delta_k)$ induces the separation used by HSD: harmonic channels represent cohomological degrees of freedom such as conserved circulation and flux, while exact and coexact channels capture metric-dependent local variation.

We describe the Global Topology (Base Space): a low-frequency harmonic forms captured efficiently in the spectral domain, avoiding costly spatial long-range computations in Section 3.2; and the Local Geometry (Ambient Fiber Space): a High-frequency gradient and curl components processed via spatial convolution, exploiting local dependencies to avoid full-graph redundancy in Section 3.3. Continuous physical models and PDE operators appear in the problem description; discrete exterior calculus, Hodge Laplacian, and tangent bundle definitions are in Appendix B; complexity analysis and implementation details are in Appendix F.

## 3.1. Discrete Operator Learning and Hodge Spectral Decomposition

Let $(\mathcal{M}, g)$ be a compact oriented Riemannian manifold with boundary, $K$ an oriented simplicial complex approximating it, and $C^k(K, \mathbb{R})$ the space of $k$-th order discrete differential forms. Denote

$$\boldsymbol{\omega}_k \in C^k(K, \mathbb{R}), \qquad \boldsymbol{f}_k \in C^k(K, \mathbb{R})$$

as the discrete unknown field and right-hand side. Given discretization $\mathbf{A}_k : C^k(K, \mathbb{R}) \to C^k(K, \mathbb{R})$ of continuous operator $\mathcal{A}^k$, the steady-state equation is

$$\mathbf{A}_k \boldsymbol{\omega}_k^\star = \boldsymbol{f}_k, \qquad \boldsymbol{\omega}_k^\star = \mathcal{G}^k(\boldsymbol{f}_k), \tag{1}$$

where $\mathcal{G}^k : C^k(K, \mathbb{R}) \to C^k(K, \mathbb{R})$ is the true solution operator. For time-dependent tasks, the same notation denotes the evolution map $\mathcal{G}_{\Delta t}^k(\boldsymbol{\omega}_k(t), \boldsymbol{f}_k(t), \boldsymbol{b}_k(t), \kappa) \mapsto \boldsymbol{\omega}_k(t + \Delta t)$, with the harmonic class determined by the initial state and boundary data. Our neural operator $\mathcal{G}_\theta^k$ approximates $\mathcal{G}^k$ directly on $C^k(K, \mathbb{R})$.

Discrete exterior calculus gives the $k$-th order Hodge–de Rham Laplacian $\mathbf{L}_k : C^k(K, \mathbb{R}) \to C^k(K, \mathbb{R})$, assembled from boundary operators, discrete exterior derivative $d_k$, codifferential $\delta_k$, and Hodge star $*_k$ (Appendix B). Construction can use topological ML libraries TopoX/TopoNetX (Hajij et al., 2022; 2024). Offline, we solve the sparse eigenvalue problem

$$\mathbf{L}_k \boldsymbol{\Psi}_k = \boldsymbol{\Psi}_k \boldsymbol{\Lambda}_k, \tag{2}$$

truncating to $m_k$ eigenpairs (all harmonic modes plus lowest-frequency non-harmonic modes), yielding orthogonalized spectral basis $\boldsymbol{\Phi}_k \in \mathbb{R}^{N_k \times m_k}$. This defines base

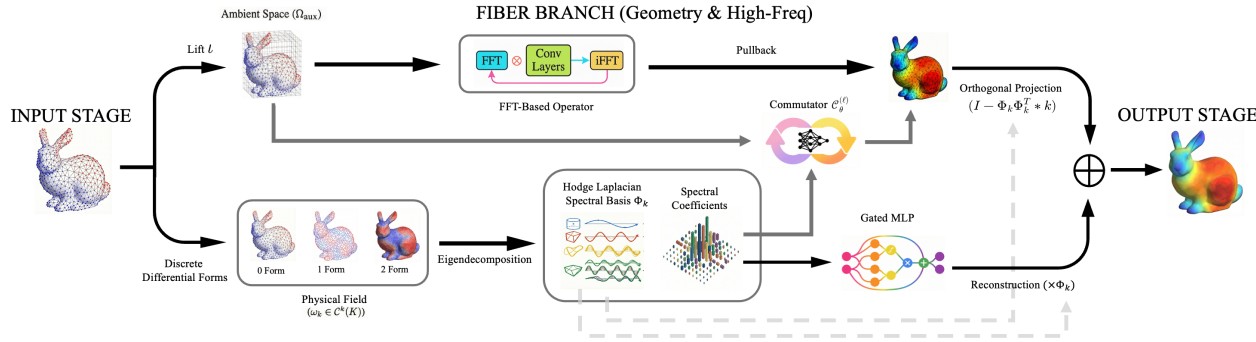

*Figure 2.* Overview of the Hodge Spectral Duality (HSD) architecture. The HSD architecture separates operator learning into a spectral Base branch (bottom) for global topology and an ambient Fiber branch (top) for high-frequency geometry. A commutator module and orthogonal projection integrate these components, ensuring strict preservation of topological invariants on manifolds.

space $\mathcal{V}_{\text{base}}^k = \text{span}(\boldsymbol{\Phi}_k)$ and orthogonal complement fiber space $\mathcal{V}_{\text{fiber}}^k$ under Hodge inner product $\langle \cdot, \cdot \rangle_{*_k}$. Projection operators and decomposition details are in Appendix C.

Fields $\boldsymbol{\omega}_k$ are decomposed via Hodge orthogonal projection: base space components (low-dimensional spectral coefficients) and fiber components (high-frequency, metric-dominated local structures). Two complementary branches model these:

$$\mathcal{G}_\theta^k = \mathcal{G}_{\text{base},\theta}^k + \mathcal{G}_{\text{fiber},\theta}^k. \qquad (3)$$

Here $\mathcal{G}_{\text{base},\theta}^k : C^k(K, \mathbb{R}) \rightarrow \mathcal{V}_{\text{base}}^k$ learns topology-dominated low-frequency response in truncated Hodge spectral domain, preserving cohomological information and conservation laws; $\mathcal{G}_{\text{fiber},\theta}^k : C^k(K, \mathbb{R}) \rightarrow \mathcal{V}_{\text{fiber}}^k$ captures metric-related high-frequency corrections via tangent bundle embedding (Appendices B, C).

### 3.2. Base Space Branch: Spectral Domain Coefficient Learning

The base space branch $\mathcal{G}_{\text{base},\theta}^k$ operates within the truncated spectral subspace $\mathcal{V}_{\text{base}}^k$: project discrete fields to Hodge spectral domain, perform physically constrained nonlinear mapping in this low-dimensional space, reconstruct to base space, achieving resolution-independent operator approximation while preserving topological structure.

At layer $\ell$, the current field $\boldsymbol{\omega}_k^{(\ell)} \in C^k(K, \mathbb{R})$ yields spectral coefficients via Hodge inner product:

$$\mathbf{c}_k^{(\ell)} = \boldsymbol{\Phi}_k^\top *_k \boldsymbol{\omega}_k^{(\ell)} \in \mathbb{R}^{m_k} \qquad (4)$$

where $\boldsymbol{\Phi}_k$ is the truncated spectral basis from equation (2) formalized in Appendix C. The coefficient vector $\mathbf{c}_k^{(\ell)}$ maintains dimension $m_k$ across mesh resolutions, encoding harmonic and low-frequency non-harmonic modes.

To embed discrete differential structure, the DEC operators $d_k$ and $\delta_k$ are pre-projected onto the truncated basis,

yielding spectral derivative matrices $\mathcal{M}_d^{(k)}$ and $\mathcal{M}_\delta^{(k)}$ (the discrete matrix forms are detailed in Appendix B). The branch constructs combined features:

$$\mathbf{q}_k^{(\ell)} = \text{concat}\big(\mathbf{c}_k^{(\ell)}, \mathcal{M}_d^{(k)} \mathbf{c}_k^{(\ell)}, \mathcal{M}_\delta^{(k)} \mathbf{c}_k^{(\ell)}\big).$$

For a 1-form input (e.g., velocity $\boldsymbol{\omega}_1$), $\mathcal{M}_d^{(1)} \mathbf{c}_1^{(\ell)}$ generates the spectral representation under exterior derivative and $\mathcal{M}_\delta^{(1)} \mathbf{c}_1^{(\ell)}$ generates the spectral representation under co-differential. These projected features are spectrally related but live in different cochain spaces: $\mathcal{M}_d^{(k)} \mathbf{c}_k^{(\ell)}$ maps the $k$-form coefficients to a $(k+1)$-form curl-type channel, while $\mathcal{M}_\delta^{(k)} \mathbf{c}_k^{(\ell)}$ maps them to a $(k-1)$-form divergence-type channel. Thus $\mathbf{q}_k^{(\ell)}$ provides explicit $(k+1)$-order (curl-type) and $(k-1)$-order (divergence-type) derivative information while updating only $k$-th order coefficients.

To capture quadratic nonlinear coupling (e.g., convection terms $\mathbf{u} \cdot \nabla\mathbf{u}$), we design a gated operator $\text{gMLP}_k$ with content and gating branches:

$$\tilde{\mathbf{c}}_k^{(\ell)} = \mathbf{W}_{\text{out}}\Big(\phi(\mathbf{W}_g \mathbf{q}_k^{(\ell)}) \odot (\mathbf{W}_c \mathbf{q}_k^{(\ell)})\Big) + \mathbf{c}_k^{(\ell)}, \qquad (5)$$

where $\mathbf{W}_g, \mathbf{W}_c, \mathbf{W}_{\text{out}}$ are learnable projections, $\phi$ is SiLU activation, and $\odot$ is Hadamard product. This gating introduces multiplicative inductive bias in spectral space, approximating nonlinear mode mixing from $\mathcal{M}_d^{(k)}$ and $\mathcal{M}_\delta^{(k)}$.

To preserve harmonic invariants in $\ker \mathbf{L}_k$, hard constraints are imposed on zero-eigenvalue modes after spectral update. When the harmonic class is specified by the initial or boundary data, HSD preserves the corresponding harmonic coefficients throughout the layer updates. Let $\mathcal{I}_H^k$ be the harmonic mode indices, then a diagonal projection $\mathbf{P}_H^k$ replaces harmonic components of $\tilde{\mathbf{c}}_k^{(\ell)}$ with reference harmonic coefficients, strictly preserving cohomology classes and global flux invariance per layer. (The definition of $\mathbf{P}_H^k$ and its relation to Betti numbers $b_k$ are provided in Appendix C.) The layer output reconstructs via $\boldsymbol{\omega}_{k,\text{base}}^{(\ell+1)} = \boldsymbol{\Phi}_k \tilde{\mathbf{c}}_k^{(\ell)}$ (complete derivation in equation (33), Appendix C), achieving

good $\mathcal{V}_{\text{base}}^k$ approximation under Hodge inner product and providing a topologically consistent low-frequency anchor for the fiber branch.

### 3.3. Fiber Branch: Metric-Dominated Correction on Tangent Bundle

The Fiber branch $\mathcal{G}_{\text{fiber},\theta}^k$ captures local high-frequency dynamics dominated by the Riemannian metric $g$ and material tensor $\kappa$ (anisotropic diffusion, boundary layers) without disrupting global topology encoded by the base branch. It models residuals in an auxiliary Euclidean spectral domain and constrains outputs to the base space complement via orthogonal projection. Mathematical consistency and Reach condition constraints are detailed in Appendix D.

We introduce structure-preserving operators between discrete form space $C^k(K, \mathbb{R}^{C_\ell})$ and an auxiliary Euclidean grid $\Omega_{\text{aux}}$. The lift operator $\iota : C^k(K) \to L^2(\Omega_{\text{aux}})$ extends discrete cochains to ambient tensor fields via Whitney forms and kernel density estimation. The pullback operator $\mathcal{R} : L^2(\Omega_{\text{aux}}) \to C^k(K)$ maps ambient fields back through trilinear interpolation and Whitney projection, forming an adjoint pair with $\iota$ under discrete Hodge inner product. In the Fiber branch, cochains are lifted to an auxiliary Euclidean field, processed by the ambient neural operator, projected back to $C^k(K)$, and restricted to the complement of the base space by $I - \Pi_{\text{base}}^k$.

Spectral convolution in ambient space uses the FNO architecture. At layer $\ell$, FFT on the auxiliary grid captures metric-related high-frequency correlations:

$$\tilde{\boldsymbol{\omega}}_{k,\text{geom}}^{(\ell)} = \mathcal{R} \circ \left( \mathcal{F}^{-1} \mathbf{R}_{\text{loc}}^{(\ell)} \mathcal{F} \right) \circ \iota \left( \boldsymbol{\omega}_k^{(\ell)} \right), \quad (6)$$

where $\mathbf{R}_{\text{loc}}^{(\ell)}$ is the learnable frequency-domain spectral kernel and $\mathcal{F}$ is FFT on $\Omega_{\text{aux}}$. This handles local geometric details via global convolution on a fixed Cartesian grid, avoiding costly anisotropic manifold convolutions.

To ensure geometric corrections preserve global conservation, we introduce orthogonal projection under the discrete Hodge inner product $\langle \boldsymbol{\alpha}, \boldsymbol{\beta} \rangle_{H^k} = \boldsymbol{\alpha}^\top \mathbf{H}_k \boldsymbol{\beta}$. Let $\Pi_{\text{base}}^k$ project onto $\mathcal{V}_{\text{base}}^k$; the Fiber output is constrained to orthogonal complement $\mathcal{V}_{\text{fiber}}^k$:

$$\boldsymbol{\omega}_{k,\text{fiber}}^{(\ell+1)} = \left( \mathbf{I} - \Pi_{\text{base}}^k \right) \tilde{\boldsymbol{\omega}}_{k,\text{geom}}^{(\ell)}. \quad (7)$$

This constraint ensures the Fiber branch corrects only high-frequency metric-dominated degrees of freedom; any low-frequency artifacts or conservation-violating modes are eliminated by projection, guaranteeing global topological invariance during cross-scale evolution.

### 3.4. Commutator Error and Spectral-Geometric Coupling

The commutator $[\mathcal{A}_{\text{Topo}}^k, \mathcal{A}_{\text{Geom}}^k] = \mathcal{A}_{\text{Topo}}^k \mathcal{A}_{\text{Geom}}^k - \mathcal{A}_{\text{Geom}}^k \mathcal{A}_{\text{Topo}}^k \neq 0$ implies operator non-commutativity: the order of applying topological and geometric operators yields different results, causing systematic splitting residuals. We introduce correction operator $\mathcal{C}_\theta^{(\ell)}$ constrained to $\mathcal{V}_k^{\text{fiber}}$ via $(\mathbf{I} - \Pi_{\text{base}}^k)$, ensuring corrections act only on high-frequency components (Appendix E).

Interaction features couple geometric lift with spectral derivatives:

$$\mathbf{z}^{(\ell)} = \iota\left(\boldsymbol{\omega}_k^{(\ell)}\right) \oplus \text{concat}\left(\mathbf{c}_k^{(\ell)}, \mathcal{M}_d^{(k)} \mathbf{c}_k^{(\ell)}, \mathcal{M}_\delta^{(k)} \mathbf{c}_k^{(\ell)}\right), \quad (8)$$

where $(\mathbf{c}_k, \mathcal{M}_d \mathbf{c}_k, \mathcal{M}_\delta \mathbf{c}_k)$ recover first-order derivatives $(d_k \boldsymbol{\omega}, \delta_k \boldsymbol{\omega})$ in discrete sense. Implementing $\mathcal{C}_\theta^{(\ell)}$ as a lightweight MLP:

$$\boldsymbol{\omega}_k^{(\ell+1)} = \mathcal{G}_{\text{base}}^{(\ell)}\left(\boldsymbol{\omega}_k^{(\ell)}\right) + (\mathbf{I} - \Pi_{\text{base}}^k)\left[\mathcal{G}_{\text{fiber}}^{(\ell)}\left(\boldsymbol{\omega}_k^{(\ell)}\right) + \mathcal{C}_\theta^{(\ell)}\left(\mathbf{z}^{(\ell)}\right)\right]. \quad (9)$$

Near-zero initialization allows gradual learning of commutator-dominated coupling from a decoupled state.

## 4. Experiments

We evaluate HSD on three tasks spanning geometric complexity, topological connectivity, and dynamic evolution: flow field reconstruction, magnetostatic field solving in multiply-connected domains, and transport processes with periodic topology; results are summarized in Table 1.

### 4.1. Baseline Models

We compare HSD against five mainstream neural operator methods. Graph Neural Operator (GNO) (Li et al., 2020b) defines kernel integration on graphs via radius neighborhood message aggregation to approximate continuous convolution on unstructured meshes. MeshGraphNets (MGN) (Pfaff et al., 2020) uses an encoder-processor-decoder architecture with MLP encoding and iterative message passing, designed for physical simulations. DeepONet (Lu et al., 2021) adopts branch-trunk decomposition: the branch encodes input function values at sensor locations, the trunk encodes query coordinates, with outputs computed via inner product. Fourier Neural Operator (FNO) (Li et al., 2020a) parameterizes kernels in the frequency domain for global dependencies; for irregular meshes, spectral convolution is performed after trilinear scattering to a Cartesian grid. Geometry-Adaptive FNO (Geo-FNO) (Li et al., 2020a) learns diffeomorphic mappings from physical coordinates to a uniform latent domain to handle geometric irregularities. All baselines are reproduced from official implementations or *torch_geometric/neuraloperator* libraries; hyperparameters are in Appendix G. We further evaluate two auxil-

*Table 1.* Comparison of main experimental results. All experiments are conducted under comparable parameter counts ($\sim$207k–310k). Best results are marked in **bold**, and second-best results are underlined. Enstrophy fidelity is not computed for the scalar field task.

| Task | Model | Params | Standard Accuracy | | Physics Consistency | | | Topological Fidelity | |
|---|---|---|---|---|---|---|---|---|---|
| | | | MSE↓ | Grad Fid↑ | Enst Fid↑ | Energy Fid↑ | Spec Fid↑ | $S_{\beta_0}$ ↑ | IoU↑ |
| External Aerodynamics | GNO | 231k | $5.40 \times 10^{-2}$ | 0.8203 | 0.6025 | 0.4328 | 0.3004 | 0.4941 | 0.2910 |
| | MGN | 247k | $5.63 \times 10^{-2}$ | 0.8599 | 0.4671 | 0.5138 | 0.4209 | 0.3982 | 0.2720 |
| | DeepONet | 239k | $2.54 \times 10^{-2}$ | 0.9448 | 0.3698 | 0.7431 | 0.6731 | 0.3625 | 0.0908 |
| | Geo-FNO | 253k | $3.20 \times 10^{-2}$ | 0.9263 | 0.2131 | 0.6438 | 0.5728 | 0.2261 | 0.0551 |
| | FNO-3D | 227k | $\underline{1.80 \times 10^{-2}}$ | $\underline{0.9663}$ | $\underline{0.6638}$ | $\underline{0.7779}$ | $\underline{0.7110}$ | $\underline{0.5584}$ | $\underline{0.3010}$ |
| | HSD (Ours) | 207k | $\mathbf{1.08 \times 10^{-2}}$ | **0.9742** | **0.7658** | **0.8906** | **0.8423** | **0.6112** | **0.3398** |
| Magneto-statics | GNO | 231k | $4.33 \times 10^{-2}$ | 0.5337 | 0.2392 | 0.3708 | 0.2373 | 0.2491 | 0.1415 |
| | MGN | 247k | $3.31 \times 10^{-3}$ | 0.9713 | 0.7016 | 0.8692 | 0.7561 | 0.5400 | 0.5653 |
| | DeepONet | 239k | $\underline{2.89 \times 10^{-4}}$ | $\underline{0.9978}$ | 0.8246 | $\underline{0.9646}$ | $\underline{0.9468}$ | $\underline{0.7877}$ | $\underline{0.7834}$ |
| | Geo-FNO | 253k | $8.97 \times 10^{-4}$ | 0.9930 | 0.8666 | 0.9044 | 0.8555 | 0.6703 | 0.7085 |
| | FNO-3D | 227k | $8.51 \times 10^{-4}$ | 0.9940 | $\underline{0.8731}$ | 0.9054 | 0.8660 | 0.7041 | 0.7236 |
| | HSD (Ours) | 212k | $\mathbf{1.84 \times 10^{-4}}$ | **0.9982** | **0.9444** | **0.9662** | **0.9492** | **0.8176** | **0.8110** |
| Toroidal Transport | GNO | 286k | $3.14 \times 10^{-2}$ | 0.5056 | — | 0.3082 | 0.0788 | 0.3795 | 0.3011 |
| | MGN | 303k | $\underline{5.23 \times 10^{-4}}$ | 0.7485 | — | 0.5232 | 0.7712 | 0.5457 | 0.5758 |
| | DeepONet | 273k | $2.48 \times 10^{-3}$ | 0.5700 | — | 0.2074 | 0.7044 | 0.3871 | 0.4433 |
| | Geo-FNO | 284k | $1.55 \times 10^{-3}$ | 0.7182 | — | 0.4104 | 0.7278 | 0.4631 | 0.5494 |
| | FNO-3D | 309k | $5.55 \times 10^{-4}$ | $\underline{0.8006}$ | — | $\underline{0.6365}$ | $\underline{0.9079}$ | $\underline{0.6721}$ | $\underline{0.7515}$ |
| | HSD (Ours) | 246k | $\mathbf{3.56 \times 10^{-4}}$ | **0.8578** | — | **0.6968** | **0.9115** | **0.7829** | **0.8131** |

*Figure 3.* Visualization of velocity vector field predictions for the External Aerodynamics task. Columns correspond to different models, with the top row showing predictions and the bottom row showing corresponding pointwise absolute errors.

iary stress tests: Ellipsoid Aero increases metric anisotropy through aspect-ratio and curvature variation, and Torus Helmholtz probes genus-one topology under oscillatory forcing. The main tables report the standard baselines shared across the three primary tasks; Appendix Table 6 gives the complementary matched-parameter comparison with recent attention/operator architectures GNOT (Hao et al., 2023), ONO (Xiao et al., 2024), and HAMLET (Bryutkin et al., 2024), where HSD obtains the lowest relative-$L^2$ error on both stress tests.

## 4.2. Evaluation Metrics

We construct a multi-dimensional evaluation framework encompassing accuracy, physical conservation, and topological consistency. Mean Squared Error (MSE) measures pointwise distance: $\text{MSE} = N^{-1} \sum_i \|u_{\text{pred}}^{(i)} - u_{\text{gt}}^{(i)}\|^2$. Gradient Fidelity (Grad Fid) evaluates gradient consistency between predicted and ground truth fields. Spectral Fidelity (Spec Fid) computes weighted relative error in the Hodge Laplacian spectral domain: $\text{Fid}_{\text{spec}} = \exp(-\alpha \|\Lambda^{-1/2}(\hat{c}_{\text{pred}} - \hat{c}_{\text{gt}})\| / \|\Lambda^{-1/2}\hat{c}_{\text{gt}}\|)$, where $\hat{c}$ denotes Hodge spectral coefficients and $\Lambda$ is the eigenvalue matrix.

Physical conservation metrics use discrete exterior differential operators. Enstrophy Fidelity (Enst Fid) compares enstrophy $\mathcal{E}_\omega = \int |\omega|^2$ (where $\omega = \nabla \times u$ is vorticity) to detect non-physical vorticity dissipation; applicable only to vector field tasks. Energy Fidelity (Energy Fid) compares Dirichlet energy $\mathcal{E}_D = \int |\nabla u|^2$, reflecting velocity field energy distribution for vector tasks and concentration gradient intensity for scalar tasks.

Topological consistency is evaluated via $\beta_0$ Score and level set IoU. The $\beta_0$ Score measures connected component consistency across thresholds: $S_{\beta_0} = \mathbb{E}_\lambda[\exp(-|\beta_0(u_{\text{pred}}^\lambda) -$

$\beta_0(u_{\text{gt}}^\lambda)|/\max(\beta_0(u_{\text{gt}}^\lambda), 1))]$, quantifying capture of independent physical features. IoU measures the isosurface geometric overlap for the spatial accuracy of high-response regions. For vector fields, topological metrics are computed on vorticity for stable features.

### 4.3. Incompressible Flow Reconstruction on Complex Geometries (External Aerodynamics)

This task simulates incompressible viscous fluid flow defined on curved manifolds (Marsden & Ratiu, 2013), with the objective of reconstructing the velocity field from vorticity distributions. The mathematical core involves solving the Laplace-Beltrami equation $\Delta_{\mathcal{M}}\psi = \zeta$ on the manifold, followed by velocity field reconstruction through orthogonal gradient decomposition $\mathbf{u} = \nabla^\perp\psi + \mathbf{u}_{\text{harm}}$ (Bhatia et al., 2012), strictly satisfying the divergence-free constraint $\nabla \cdot \mathbf{u} = 0$. The operator learning task is defined as mapping from the scalar vorticity field $\omega$ on the manifold surface to the velocity vector field $\mathbf{u}$ in the tangent space (Kovachki et al., 2023).

Geometric data are sourced from high-fidelity automotive triangular meshes in the DrivAerNet++ dataset (Elrefaie et al., 2024), sampled to 3000 nodes. This scenario exhibits high geometric complexity, containing non-smooth features, sharp edges, and regions with curvature variations. On closed surfaces, fluid dynamics are constrained by global geometric properties (Arnold et al., 2010), and the velocity solution depends on local vorticity distributions together with global circulation constraints determined by the surface genus (Hatcher, 2002). Numerical solutions employ graph discrete differential operators to approximate continuous operators on surfaces (Bronstein et al., 2017), with sparse linear solvers handling anisotropic diffusion problems (Desbrun et al., 2003).

Table 1 (upper) shows results for this task. MSE of $1.08 \times 10^{-2}$ (40% reduction vs FNO-3D). On enstrophy fidelity (0.7658) and spectral fidelity (0.8423)—key metrics for turbulent microstructures and energy distributions, HSD effectively avoids over-smoothing and captures high-frequency vortex features. Geo-FNO's coordinate deformation mechanism struggles with complex industrial geometries under constrained parameter scales, losing subtle physical features. Additional visualizations are in Appendix H.2.

### 4.4. Magnetostatic Fields in Multiply-Connected Domains (Magnetostatics)

This task simulates magnetostatic field problems excited by source distributions in three-dimensional space, with the physics described by the steady-state form of Maxwell's equations (Jackson, 2021). The magnetic flux density $\mathbf{B}$ is modeled as a linear superposition of scalar potential gradients and non-local harmonic fields induced by topological defects $\mathbf{B} = -\nabla\phi + \mathbf{B}_{\text{harm}}$, where the scalar potential satisfies the Poisson equation $\Delta\phi = \rho_m$ (Bhatia et al., 2012). The operator learning task is defined as mapping from the source scalar field (magnetic charge density $\rho_m$) to the divergence-free magnetic flux vector field $\mathbf{B}$ (Kovachki et al., 2023).

The computational domain is a three-dimensional bounding box containing spherical shell obstacles, forming a multiply-connected region. This geometric structure is discretized using unstructured tetrahedral meshes (Si, 2015), sampled to 3000 nodes. The presence of spherical shell obstacles makes the computational domain non-simply connected, and the magnetic field must contain global flux components induced by geometric cavities in addition to the irrotational component generated by local sources (Nakahara, 2003). Numerical solutions employ finite element methods to discretize the governing equations, combined with multipole expansion techniques to construct global harmonic bases satisfying boundary conditions (Jin, 2015).

Table 1 (middle) shows results for this task. Magnetostatics emphasizes global topological obstacles and long-range field coupling, where DeepONet achieves lower MSE and optimal spectral fidelity via global fitting. HSD maintains comprehensive advantages: MSE of $1.84 \times 10^{-4}$ (36% reduction vs DeepONet) and enstrophy fidelity of 0.9444, precisely identifying local flux concentration structures that DeepONet statistically smooths out, preserving energy distribution and topological features (Karniadakis et al., 2021). Additional visualizations are in Appendix H.1.

### 4.5. Advection-Diffusion Dynamics (Toroidal Transport)

This task simulates time-varying transport processes of scalar fields driven by a given velocity field (LeVeque, 2002), governed by the unsteady advection-diffusion equation $\partial u/\partial t + \nabla \cdot (\mathbf{v}u) = \nu\Delta u$. The system couples hyperbolic advective transport mechanisms with parabolic diffusive dissipation mechanisms, exhibiting significant spatiotemporal multiscale characteristics (Quarteroni, 2009). The operator learning task is defined as mapping from the initial scalar concentration distribution $u_0$ to the temporal evolution trajectory of the scalar field $u(t)$.

The computational domain is a torus embedded in three-dimensional space, topologically possessing non-zero genus (Genus=1) with two independent non-contractible closed loop paths (Hatcher, 2002), sampled to 3000 nodes. The multiply-connected nature of the torus gives the scalar field transport process a re-entrant characteristic, where physical quantities must maintain continuity after traversing around the torus, requiring the model to possess long-range spatiotemporal modeling capabilities beyond local receptive fields (Gu & Yau, 2008). Numerical solutions employ a hybrid strategy, with the advection term using finite volume

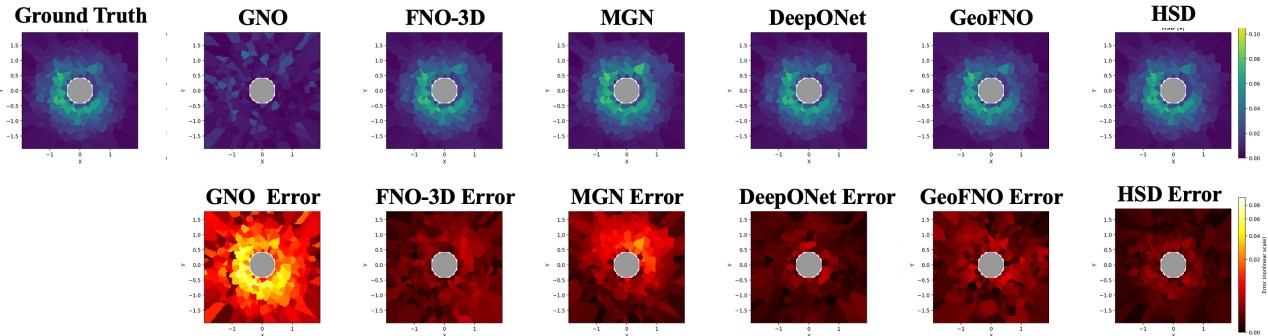

*Figure 4.* Slice visualization of magnetic vector field at $z = 0$ plane for the Magnetostatics task. Columns correspond to different models, with the top row showing predictions and the bottom row showing corresponding errors, with the color scale: black→red→yellow→white indicates increasing error.

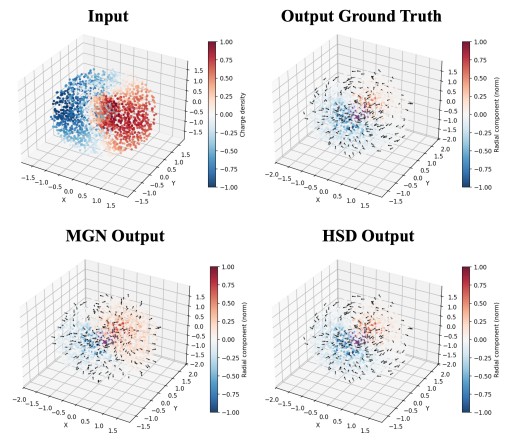

*Figure 5.* Comparison of Magnetostatics field predictions.

*Figure 6.* Topological contours: level set connectivity (threshold at 50% of range). The $\beta_0$ values for each model are: GT=3, HSD=3, GNO=2, DeepONet=0.

methods with flux limiters in upwind schemes (Versteeg & Malalasekera, 2007), and the diffusion term using implicit time integration schemes (Hairer & Wanner, 1996).

Table 1 (lower) shows results for this task. MGN achieves MSE of $5.23 \times 10^{-4}$, demonstrating message passing effectiveness for local temporal evolution. HSD achieves MSE of $3.56 \times 10^{-4}$ (36% reduction vs FNO-3D), with Energy Fidelity (0.6968 vs 0.6365) and $\beta_0$ Score (0.7829 vs 0.6721) significantly outperforming FNO-3D, indicating stable preservation of energy dissipation and topological structure under long-time integration while avoiding non-physical artifacts (Li et al., 2020a). Additional visualizations are in Appendix H.3.

*Table 2.* Ablation study results for core components (MSE). Percentages in parentheses indicate performance degradation relative to complete HSD.

| Model Variant | Magnetostatics | Ext. Aero. | Toroidal Trans. |
|---|---|---|---|
| HSD (Full) | $\mathbf{1.84 \times 10^{-4}}$ | $\mathbf{1.08 \times 10^{-2}}$ | $\mathbf{3.56 \times 10^{-4}}$ |
| w/o $\mathcal{C}_\theta$ | $2.18 \times 10^{-4}$(+18%) | $1.17 \times 10^{-2}$(+8%) | $3.79 \times 10^{-4}$(+6%) |
| w/o $\Pi_{\text{base}}$ | $2.20 \times 10^{-4}$(+20%) | $1.45 \times 10^{-2}$(+34%) | $3.72 \times 10^{-4}$(+4%) |
| FNO-3D | $8.51 \times 10^{-4}$(+363%) | $1.80 \times 10^{-2}$(+67%) | $5.55 \times 10^{-4}$(+56%) |

### 4.6. Spectral Bias Analysis

Figure 8 shows the predicted field energy distribution on Hodge Laplacian eigenmodes. DeepONet, Geo-FNO, and GNO exhibit rapid energy decay with increasing eigenfrequency, falling below ground truth, indicating high-frequency filtering. HSD's spectrum closely matches the ground truth in high-frequency regions, overcoming spectral bias. The Fiber branch effectively compensates the Base branch's high-frequency deficiencies, validating the spectral-geometric duality design.

### 4.7. Training Efficiency and Computational Complexity

Despite introducing spectral decomposition preprocessing, HSD maintains significant training efficiency advantages. The key is decoupling offline geometric encoding from online learning: Hodge Laplacian eigendecomposition executes once per mesh, taking approximately 57 seconds on the most complex tetrahedral mesh (Magnetostatics, ~20k elements) and only seconds on surface meshes.

Online complexity is $\mathcal{O}(Nk)$ for spectral projection and $\mathcal{O}(N \log N)$ for FFT, versus message-passing MGN's $\mathcal{O}(N|E|)$. Including all preprocessing, HSD's total training time is 56× faster than MGN in External Aerodynamics (33s vs 1865s) and only 5% of MGN in Magnetostatics (215s vs 3983s), while remaining comparable to purely Euclidean FNO-3D. Complete comparisons are in Appendix Table 7.

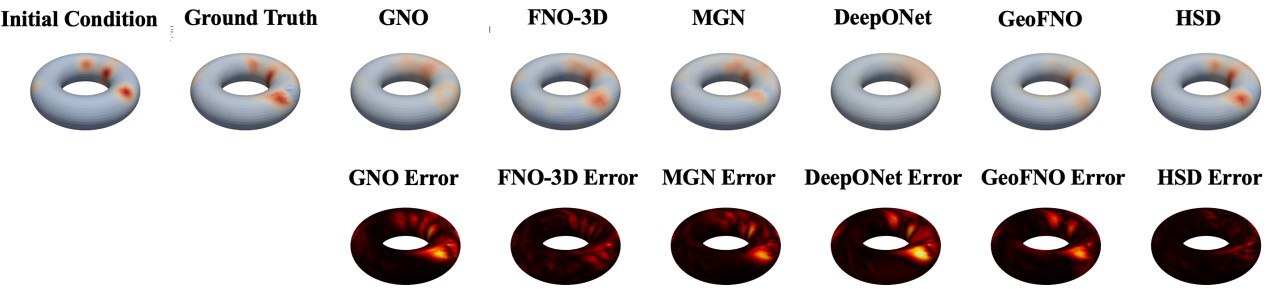

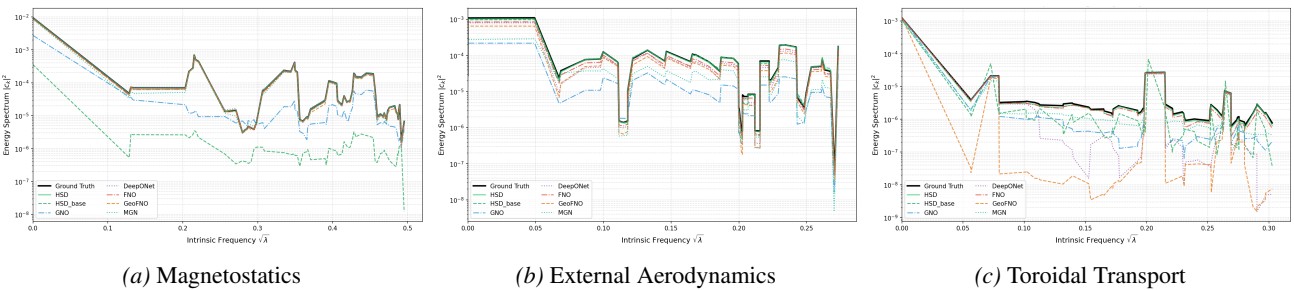

*Figure 7.* Comparison of initial conditions, final ground truth field, and scalar field predictions from each model. Columns correspond to different models, with the top row showing predicted fields and the bottom row showing corresponding errors relative to Ground Truth, with the same color scale as Figure 4.

*(a)* Magnetostatics  *(b)* External Aerodynamics  *(c)* Toroidal Transport

*Figure 8.* Spectral energy decay analysis of predicted fields across the three tasks. The horizontal axis represents eigenfrequency $\lambda$, and the vertical axis represents spectral coefficient energy $|c_k|^2$.

*Table 3.* Impact of spectral truncation number $k$ on model performance (MSE). Percentages in parentheses indicate change relative to $k = 64$.

| Modes $k$ | Magnetostatics | Ext. Aero. | Toroidal Trans. |
|---|---|---|---|
| $k = 64$ | $1.84 \times 10^{-4}$ | $1.08 \times 10^{-2}$ | $3.56 \times 10^{-4}$ |
| $k = 128$ | $1.64 \times 10^{-4}(\downarrow 11\%)$ | $9.32 \times 10^{-3}(\downarrow 14\%)$ | $2.80 \times 10^{-4}(\downarrow 21\%)$ |
| $k = 256$ | $1.58 \times 10^{-4}(\downarrow 14\%)$ | $8.99 \times 10^{-3}(\downarrow 17\%)$ | $2.73 \times 10^{-4}(\downarrow 23\%)$ |

### 4.8. Ablation Studies

Table 2 analyzes core component contributions. Removing orthogonal projection causes spectral convolution to introduce non-physical low-frequency noise, with the largest degradation on geometrically complex domains. Removing the commutator MLP most impacts multiply-connected domains, confirming that topological-geometric operator non-commutativity requires compensation via $\mathcal{C}_\theta$.

Table 3 shows that increasing spectral modes improves performance with diminishing returns, validating the spectral-geometric duality: the Base branch needs only a few low-frequency modes for global topology, while the Fiber branch efficiently captures high-frequency details without costly large eigenbases.

On External Aerodynamics, HSD exhibits stable generalization under remeshing: when the inference mesh density increases from 3000 to 7000 nodes, its error varies by at most 30%, whereas all baselines suffer at least a $10\times$ larger error amplification, indicating HSD learns the underlying physical operator across mesh resolutions.

### 5. Discussion, Limitations and Scope

This method relies on sparse spectral decomposition of the discrete Hodge Laplacian, requiring fixed geometry, iso-morphic/isometric deformations, or minor perturbations to amortize eigendecomposition costs to offline precomputation. The framework is thus currently suited for Eulerian-perspective simulations on manifolds of three dimensions or less. For scenarios requiring per-step mesh topology reconstruction, future work will incorporate iso-spectral deformation or Functional Maps theory to enable low-cost spectral basis transfer across time-varying geometry and non-isometric deformations without repeated eigendecomposition. Additionally, the low-pass characteristics of ambient mollification accommodate high-gradient continuous structures (e.g., boundary layers) but not strong discontinuities such as shock waves, restricting applicability to shock-free regimes. For driven time-dependent PDEs, harmonic coefficients associated with injected sources may require task-specific updates. Appendix G.2 further examines mesh, graph, and point-cloud inputs on the auxiliary stress tests, and separately reports a Betti-number prediction study that probes topology recognition from Hodge-based features. Overall, the architecture captures high-frequency geometric details together with global conservation laws on complex manifolds, supporting an intrinsic Hodge-orthogonal additive structure for operator learning on manifolds.

## Impact Statement

This paper presents work whose goal is to advance the field of Machine Learning. There are many potential societal consequences of our work, none of which we feel must be specifically highlighted here.

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

# A. Notation Reference Table and A Primer on Algebraic Topology

| Symbol | Type | Meaning |
|---|---|---|
| $(\mathcal{M}, g)$ | Continuous geometry | Compact oriented Riemannian manifold with boundary and its Riemannian metric. |
| $\partial\mathcal{M}$ | Continuous geometry | Boundary of manifold $\mathcal{M}$. |
| $K$ | Discrete geometry | Oriented simplicial complex approximating $(\mathcal{M}, g)$. |
| $K_k,\ N_k = |K_k|$ | Discrete geometry | Set of $k$-dimensional simplices and its cardinality. |
| $C^k(K, \mathbb{R})$ | Space | Space of $k$-th order discrete differential forms ($k$-cochains), carrying discrete degrees of freedom of $k$-th order physical quantities. |
| $\boldsymbol{\omega}_k \in C^k(K, \mathbb{R})$ | State | Discrete representation of unknown $k$-th order physical field. |
| $\boldsymbol{f}_k \in C^k(K, \mathbb{R})$ | Data | Discrete representation of right-hand side/source term or external force. |
| $\mathcal{A}^k$ | Continuous operator | Control operator acting on $k$-th order differential forms in continuous PDE. |
| $\mathbf{A}_k : C^k \to C^k$ | Discrete operator | Discretization of $\mathcal{A}^k$ on $C^k(K, \mathbb{R})$. |
| $\mathcal{G}^k$ | Solution operator | (True) solution operator satisfying $\boldsymbol{\omega}_k^\star = \mathcal{G}^k(\boldsymbol{f}_k)$. |
| $\mathcal{G}_\theta^k$ | Neural operator | Neural operator with parameters $\theta$, used to approximate $\mathcal{G}^k$. |
| $d,\ \delta$ | Continuous operator | Exterior derivative $d$ and codifferential $\delta$. |
| $\Delta_k = d\delta + \delta d$ | Continuous operator | $k$-th order Hodge–de Rham Laplacian. |
| $\mathbf{B}_k$ | DEC operator | Oriented boundary matrix between $k$-dimensional and $(k-1)$-dimensional simplices. |
| $d_k = \mathbf{B}_{k+1}^\top$ | DEC operator | Discrete exterior derivative $C^k(K, \mathbb{R}) \to C^{k+1}(K, \mathbb{R})$. |
| $*_k$ | DEC operator | $k$-th order discrete Hodge star operator (mass matrix), inducing discrete inner product. |
| $\langle \boldsymbol{\alpha}, \boldsymbol{\beta} \rangle_{*_k} = \boldsymbol{\alpha}^\top *_k \boldsymbol{\beta}$ | Inner product | Discrete Hodge inner product on $C^k(K, \mathbb{R})$. |
| $\delta_k = *_{k-1}^{-1} \mathbf{B}_k *_k$ | DEC operator | Discrete codifferential $C^k(K, \mathbb{R}) \to C^{k-1}(K, \mathbb{R})$. |
| $\mathbf{L}_k = d_{k-1}\delta_k + \delta_{k+1}d_k$ | DEC operator | $k$-th order discrete Hodge Laplacian. |
| $b_k = \dim \ker \mathbf{L}_k$ | Topological quantity | $k$-th Betti number, corresponding to the dimension of harmonic $k$-form space. |
| $\mathbf{L}_k \boldsymbol{\Psi}_k = \boldsymbol{\Psi}_k \boldsymbol{\Lambda}_k$ | Spectral decomposition | Eigendecomposition of discrete Hodge Laplacian. |
| $\psi_{k,i}$ | Mode | $i$-th eigenvector of $\mathbf{L}_k$ ($k$-th order spectral mode). |
| $\lambda_{k,i}$ | Eigenvalue | $i$-th eigenvalue of $\mathbf{L}_k$. |
| $\boldsymbol{\Psi}_k$ | Matrix | Matrix composed of all eigenvectors $\psi_{k,i}$. |
| $\boldsymbol{\Lambda}_k$ | Matrix | Diagonal eigenvalue matrix $\mathrm{diag}(\lambda_{k,1}, \ldots, \lambda_{k,N_k})$. |
| $m_k$ | Truncation dimension | Number of modes corresponding to the smallest eigenvalues retained in truncated spectrum (including all harmonic modes). |
| $\boldsymbol{\Phi}_k \in \mathbb{R}^{N_k \times m_k}$ | Basis | Truncated spectral basis $[\psi_{k,1}, \ldots, \psi_{k,m_k}]$, orthogonalized under $*_k$. |
| $\mathcal{V}_{\mathrm{base}}^k = \mathrm{span}(\boldsymbol{\Phi}_k)$ | Subspace | Base space spectral subspace containing harmonic modes and several low-frequency modes. |
| $\mathcal{V}_{\mathrm{fiber}}^k = (\mathcal{V}_{\mathrm{base}}^k)^{\perp *_k}$ | Subspace | Hodge orthogonal complement of $\mathcal{V}_{\mathrm{base}}^k$, high-frequency / local degree of freedom subspace. |
| $\Pi_{\mathrm{base}}^k = \boldsymbol{\Phi}_k \boldsymbol{\Phi}_k^\top *_k$ | Projection | Orthogonal projection operator onto $\mathcal{V}_{\mathrm{base}}^k$. |
| $\boldsymbol{\omega}_{k,\mathrm{base}} = \Pi_{\mathrm{base}}^k \boldsymbol{\omega}_k$ | Component | Base space (topological + low-frequency) component of $\boldsymbol{\omega}_k$. |
| $\boldsymbol{\omega}_{k,\mathrm{fiber}} = (\mathbf{I} - \Pi_{\mathrm{base}}^k)\boldsymbol{\omega}_k$ | Component | Fiber (high-frequency / metric-dominated) component of $\boldsymbol{\omega}_k$. |
| $\mathcal{A}_{\mathrm{Topo}}^k$ | Continuous operator | Topology-dominated operator component generated by $d, \delta, \Delta_k$. |
| $\mathcal{A}_{\mathrm{Geom}}^k$ | Continuous operator | Geometry / material-dominated operator component depending on $g, \kappa$. |
| $\mathcal{G}_{\mathrm{base}}^k : C^k \to \mathcal{V}_{\mathrm{base}}^k$ | Solution operator | Solution operator mapping input to base space subspace. |
| $\mathcal{G}_{\mathrm{fiber}}^k : C^k \to \mathcal{V}_{\mathrm{fiber}}^k$ | Solution operator | Solution operator mapping input to fiber subspace. |
| $\mathcal{G}_{\mathrm{base},\theta}^k,\ \mathcal{G}_{\mathrm{fiber},\theta}^k$ | Neural operator | Learnable approximations of $\mathcal{G}_{\mathrm{base}}^k$ and $\mathcal{G}_{\mathrm{fiber}}^k$. |
| $\boldsymbol{\omega}_k^{(\ell)}$ | Network state | Current field approximation at layer $\ell$. |
| $\mathbf{c}_k^{(\ell)} = \boldsymbol{\Phi}_k^\top *_k \boldsymbol{\omega}_k^{(\ell)}$ | Spectral coefficients | Coordinates of $\boldsymbol{\omega}_k^{(\ell)}$ under truncated spectral basis. |
| $\mathcal{M}_d^{(k)} = \boldsymbol{\Phi}_{k+1}^\top *_{k+1} d_k \boldsymbol{\Phi}_k$ | Matrix | Discrete exterior derivative operator under truncated spectral basis. |
| $\mathcal{M}_\delta^{(k)} = \boldsymbol{\Phi}_{k-1}^\top *_{k-1} \delta_k \boldsymbol{\Phi}_k$ | Matrix | Discrete codifferential operator under truncated spectral basis. |
| $\mathbf{q}_k^{(\ell)}$ | Features | Spectral domain features: $\mathrm{concat}(\mathbf{c}_k^{(\ell)}, \mathcal{M}_d^{(k)} \mathbf{c}_k^{(\ell)}, \mathcal{M}_\delta^{(k)} \mathbf{c}_k^{(\ell)})$. |
| $\mathrm{MLP}_k$ | Network | Small multilayer perceptron acting on $\mathbf{q}_k^{(\ell)}$. |
| $\tilde{\mathbf{c}}_k^{(\ell)}$ | Spectral coefficients | Spectral coefficients after update by $\mathrm{MLP}_k$. |
| $\mathcal{I}_H^k$ | Index set | Spectral index set corresponding to zero eigenvalues (harmonic modes). |
| $\mathbf{P}_H^k$ | Projection | Diagonal projection matrix onto harmonic subspace. |

| Symbol | Type | Meaning |
| --- | --- | --- |
| $\boldsymbol{\omega}_{k,\text{base}}^{(\ell+1)} = \boldsymbol{\Phi}_k \tilde{\mathbf{c}}_k^{(\ell)}$ | Component | Field component reconstructed from base space branch at layer $\ell+1$. |
| $\iota$ | Lift operator | Lifts discrete forms to continuous fields on a manifold/auxiliary Euclidean domain (Whitney forms + kernel smoothing). |
| $\mathcal{R}$ | Pullback operator | Interpolates / projects ambient fields back to $C^k(K, \mathbb{R})$. |
| $\Omega_{\text{aux}}$ | Domain | Auxiliary Euclidean domain surrounding the manifold (domain of regular voxel grid). |
| $\mathcal{F}, \mathcal{F}^{-1}$ | Operator | Discrete fast Fourier transform and its inverse on $\Omega_{\text{aux}}$. |
| $\mathbf{R}_{\text{loc}}^{(\ell)}$ | Kernel | Frequency domain convolution kernel tensor for ambient FNO at layer $\ell$. |
| $\tilde{\boldsymbol{\omega}}_{k,\text{geom}}^{(\ell)}$ | Field | Geometric residual field after ambient FNO correction. |
| $\boldsymbol{\omega}_{k,\text{fiber}}^{(\ell+1)} = (\mathbf{I} - \Pi_{\text{base}}^k)\tilde{\boldsymbol{\omega}}_{k,\text{geom}}^{(\ell)}$ | Component | Fiber branch output after projection onto $\mathcal{V}_{\text{fiber}}^k$. |
| $\mathcal{C}_\theta^{(\ell)}$ | Correction operator | Learnable correction term approximating commutator error $[\mathcal{A}_{\text{Topo}}^k, \mathcal{A}_{\text{Geom}}^k]$. |
| $\mathbf{z}^{(\ell)}$ | Features | Spectral-geometric interaction features (concatenation of $\iota(\boldsymbol{\omega}_k^{(\ell)})$ and $\mathbf{q}_k^{(\ell)}$). |
| $\lambda$ | Scalar | Global scaling/confidence coefficient for Fiber geometric residual (ResScale). |
| $\tau_{\mathcal{M}}$ | Geometric quantity | Reach of manifold $\mathcal{M}$, controlling geometric safety radius of ambient embedding. |
| $h, h_{\text{aux}}$ | Scale | Simplicial complex mesh scale and auxiliary voxel grid step size. |
| $\epsilon$ | Scale | Ambient kernel smoothing bandwidth (mollification scale). |
| $\gamma$ | Mesh quality | Maximum condition number of element geometric matrices, characterizing mesh anisotropy. |

To facilitate the understanding of the Hodge Spectral Duality framework within the machine learning community, this section reformulates the core concepts of Discrete Exterior Calculus (DEC) and Algebraic Topology using standard linear algebra and graph signal processing terminology. We emphasize exact mathematical definitions over heuristic metaphors.

## A.1. Data Representation: From Node Signals to Cochains

In standard Graph Neural Networks (GNNs), data is typically treated as signals on vertices ($X \in \mathbb{R}^{|V| \times c}$). In our framework, physical fields are strictly typed by their integration domains, formalized as **Discrete Differential Forms** (or Cochains).

- **Linear Algebra Perspective**: The space of $k$-forms, denoted as $C^k(K, \mathbb{R})$, is simply a vector space $\mathbb{R}^{N_k}$, where $N_k$ is the number of $k$-dimensional simplices (vertices, edges, faces).
  - **0-forms** ($u \in \mathbb{R}^{N_0}$)**:** Scalars defined on vertices (e.g., Temperature).
  - **1-forms** ($v \in \mathbb{R}^{N_1}$)**:** Scalars defined on oriented edges (e.g., Flow rate along a pipe). Note that changing edge orientation negates the value.
  - **2-forms** ($w \in \mathbb{R}^{N_2}$)**:** Scalars defined on oriented faces (e.g., Flux through a surface element).
- **Insight for ML**: Unlike GNNs, which learn arbitrary feature vectors, this framework enforces a strict *dimension-binding inductive bias*. A velocity field must be processed as a 1-form (edge signal), not a 0-form (node feature), to preserve its transformation properties under geometric deformation.

## A.2. The Algebraic Structure: Boundary and Coboundary

The core connectivity of the mesh is encoded in the **Boundary Operator** $\partial_k$.

- **Definition**: $\partial_k$ maps a $k$-simplex to a linear combination of its $(k-1)$-faces.
- **Matrix Representation**: In the discrete setting, this is exactly the incidence matrix $B_k \in \mathbb{R}^{N_{k-1} \times N_k}$. For example, for an edge $e_{ij} = [v_i, v_j]$, the boundary is $v_j - v_i$. Thus, $B_1$ is the standard edge-to-vertex incidence matrix containing only $\{0, 1, -1\}$.

The **Exterior Derivative** $d_k$ (used for Gradient, Curl) is the **adjoint** (transpose) of the boundary operator:

$$d_k = B_{k+1}^\top \tag{10}$$

- $d_0$ **(Gradient)**: Maps vertex signals to edge signals (computes differences).

- $d_1$ **(Curl)**: Maps edge signals to face signals (sums circulation around a face).

**The Fundamental Property**: The defining algebraic structure of a complex is $\partial_{k-1} \circ \partial_k = 0$. In matrix terms:

$$B_k B_{k+1} = \mathbf{0} \quad \implies \quad d_{k+1} d_k = \mathbf{0} \tag{11}$$

This identity ($d^2 = 0$) strictly guarantees that "the Curl of a Gradient is zero" and "the Divergence of a Curl is zero" at machine precision purely via sparse matrix multiplication, without needing to learn these physics constraints.

### A.3. The Generalized Laplacian and Hodge Decomposition

Standard GCNs utilize the Graph Laplacian $L_0 = D - A \approx B_1^\top B_1$. Hodge Theory generalizes this to higher dimensions:

$$L_k = \underbrace{d_{k-1}\delta_k}_{\text{Grad-Div term}} + \underbrace{\delta_{k+1}d_k}_{\text{Curl-Curl term}} \tag{12}$$

- **Spectral Interpretation**: $L_k$ is a symmetric positive semi-definite matrix. Its eigenvectors provide a Fourier basis for signals on edges ($k = 1$) or faces ($k = 2$).

- **Hodge Decomposition**: Just as any vector can be projected onto orthogonal axes, any discrete field $\omega \in \mathbb{R}^{N_k}$ decomposes orthogonally into three subspaces determined by the operators above:

$$\omega = \text{im}(d_{k-1}) \oplus \text{im}(\delta_{k+1}) \oplus \ker(L_k) \tag{13}$$

This separates the signal into **Irrotational** (gradient-flow), **Solenoidal** (divergence-free), and **Harmonic** components.

### A.4. Betti Numbers: Topological Invariants as Null Spaces

**Betti numbers** ($b_k$) are often cited abstractly, but in our computational framework, they have a precise linear algebraic definition related to the **Cohomology Groups**.

- **Definition**: The $k$-th Betti number is the dimension of the kernel (null space) of the $k$-th Hodge Laplacian.

$$b_k = \dim(\ker(L_k)) = \text{number of zero eigenvalues of } L_k \tag{14}$$

- **Physical Meaning in $\mathbb{R}^3$**:
    - $b_0$: Number of connected components. A harmonic 0-form is constant on each component.
    - $b_1$: Number of independent non-contractible loops (e.g., flow circulating around a handle or hole). A harmonic 1-form represents a circulation that cannot be explained by a local gradient potential.
    - $b_2$: Number of enclosed voids (cavities). A harmonic 2-form represents flux trapped on a closed surface.

**Relevance to Operator Learning**: In standard neural networks, global topological features (like $b_1$ circulation) often get smoothed out by local message passing. By explicitly projecting onto the kernel of $L_k$ (the harmonic subspace), our method preserves these global invariants as hard constraints, ensuring the network respects the fundamental topology of the physical domain.

## B. Mathematical Foundations of Discrete Exterior Calculus and Tangent Bundle

This appendix provides the strict mathematical definitions of discrete exterior calculus, Hodge spectral structure, and tangent bundle, supporting the construction of the Hodge Spectral Duality neural operator in the main text. For related theory, see (Hirani, 2003; Desbrun et al., 2003; Arnold et al., 2006; 2010).

## B.1. Simplicial Complex and Discrete Differential Forms

Let $(\mathcal{M}, g)$ be a compact oriented $n$-dimensional Riemannian manifold with boundary. To approximate $(\mathcal{M}, g)$ on a computer, take an oriented simplicial complex embedded in Euclidean space

$$K = (V, E, F, \dots),$$

where $V, E, F$ are the sets of vertices, edges, and faces respectively, corresponding to $0, 1, 2$-dimensional simplices. In general, denote $K_k$ as the set of $k$-dimensional simplices and $N_k = |K_k|$ as its cardinality. Cases where $k > n$ are not considered.

The space of discrete $k$-th order differential forms $C^k(K, \mathbb{R})$ is defined as the real vector space of all mappings from each $k$-dimensional simplex $\sigma \in K_k$ to a real number:

$$C^k(K, \mathbb{R}) \simeq \mathbb{R}^{N_k},$$

where each component corresponds to an integral quantity on a $k$-dimensional simplex. $C^0(K, \mathbb{R})$ corresponds to discrete scalar fields on vertices, $C^1(K, \mathbb{R})$ corresponds to line integral fluxes on oriented edges, $C^2(K, \mathbb{R})$ corresponds to area fluxes or area densities on oriented faces, and higher-order spaces $C^k(K, \mathbb{R})$ follow analogously.

To represent adjacency and orientation relationships between simplices, for $k \geq 1$, define the oriented boundary matrix

$$\mathbf{B}_k \in \mathbb{R}^{N_{k-1} \times N_k},$$

whose $(i, j)$-th component is

$$[\mathbf{B}_k]_{ij} = \begin{cases} 0, & \sigma_{k-1}^i \not\subset \partial\sigma_k^j, \\ \pm 1, & \sigma_{k-1}^i \subset \partial\sigma_k^j, \end{cases}$$

where the sign is determined by the relative orientation between simplices. Matrix $\mathbf{B}_k$ encodes the oriented $(k-1)$-dimensional boundary of each $k$-dimensional simplex.

## B.2. Discrete Exterior Derivative, Hodge Star, and Codifferential

The discrete counterpart of the continuous exterior derivative operator $d : \Omega^k(\mathcal{M}) \to \Omega^{k+1}(\mathcal{M})$ is determined by the algebraic properties of the boundary operator. Define the discrete exterior derivative

$$d_k : C^k(K, \mathbb{R}) \to C^{k+1}(K, \mathbb{R})$$

at the matrix level as

$$d_k = \mathbf{B}_{k+1}^\top. \tag{15}$$

For $k = 0$, $d_0$ gives the discrete gradient operator on graph vertices; for $k = 1$, $d_1$ gives a discrete curl-type operator on edges. From $\mathbf{B}_k \mathbf{B}_{k+1} = 0$ we obtain $d_{k+1} d_k = 0$, reflecting the complex structure of the discrete exterior derivative.

To introduce a metric-consistent inner product between discrete forms, define the discrete Hodge star operator

$$*_k : C^k(K, \mathbb{R}) \to C^k(K, \mathbb{R}),$$

The continuous Hodge star maps $\Omega^k(\mathcal{M})$ to $\Omega^{n-k}(\mathcal{M})$; the discrete object $*_k$ used here is its matrix representation after pairing primal and dual cells, equivalently the mass matrix that induces the discrete $L^2$ Hodge inner product on $C^k(K, \mathbb{R})$. Given $*_k$, introduce the inner product on $C^k(K, \mathbb{R})$

$$\langle \boldsymbol{\alpha}, \boldsymbol{\beta} \rangle_{*_k} = \boldsymbol{\alpha}^\top *_k \boldsymbol{\beta}, \qquad \boldsymbol{\alpha}, \boldsymbol{\beta} \in C^k(K, \mathbb{R}). \tag{16}$$

This inner product approximates at the mesh level the continuous $L^2$ inner product

$$\langle \alpha, \beta \rangle = \int_{\mathcal{M}} \alpha \wedge *\beta.$$

The discrete codifferential operator

$$\delta_k : C^k(K, \mathbb{R}) \to C^{k-1}(K, \mathbb{R})$$

is defined as the formal adjoint of $d_{k-1}$ with respect to inner product (16), i.e.,

$$\delta_k = *_{k-1}^{-1} \mathbf{B}_k *_k, \tag{17}$$

satisfying $\delta_k^2 = 0$. For $k = 1$, $\delta_1$ corresponds to the discrete divergence operator; for $k = 2$, $\delta_2$ corresponds to higher-order divergence. Similar to the continuous case, the combination of $d_k$ and $\delta_k$ uniformly describes discrete versions of first-order differential operators such as gradient, curl, and divergence.

Based on the above operators, the discrete Hodge–de Rham Laplacian is defined as

$$\mathbf{L}_k = d_{k-1}\delta_k + \delta_{k+1}d_k : C^k(K, \mathbb{R}) \to C^k(K, \mathbb{R}), \tag{18}$$

whose matrix is symmetric positive semi-definite. For $k = 0$, if selecting standard volume metric with $*_0 = \mathbf{I}$, then

$$\mathbf{L}_0 = \mathbf{B}_1^\top \mathbf{B}_1,$$

which is the combinatorial graph Laplacian (unnormalized form) commonly used in graph learning. For general $k$, $\mathbf{L}_k$ generalizes the graph Laplacian to generalized Laplacian operators on higher-order cells such as edges and faces.

## B.3. Hodge–de Rham Decomposition and Hodge Spectrum

The continuous Hodge–de Rham decomposition states that, under appropriate boundary conditions, each smooth $k$-form can be uniquely decomposed into three parts: gradient-type, curl/divergence-type, and harmonic-type. Discrete exterior calculus and finite element exterior calculus theory show that, on appropriate discrete shape function spaces, this decomposition still holds at the discrete level (Hirani, 2003; Desbrun et al., 2003; Arnold et al., 2006; 2010). Specifically, on $C^k(K, \mathbb{R})$ there is an orthogonal decomposition

$$C^k(K, \mathbb{R}) = \operatorname{im} d_{k-1} \ \oplus \ \operatorname{im} \delta_{k+1} \ \oplus \ \ker \mathbf{L}_k, \tag{19}$$

where $\operatorname{im} d_{k-1}$ is the gradient-type component, describing parts driven by scalar potentials; $\operatorname{im} \delta_{k+1}$ is the curl-type or divergence-type component, describing parts driven by circulation or sources/sinks; $\ker \mathbf{L}_k$ is the harmonic subspace, describing modes that are locally source-free but constrained by global topology.

To characterize the multi-scale structure and topological modes of $k$-th order fields, consider the spectral decomposition of the discrete Hodge Laplacian

$$\mathbf{L}_k \boldsymbol{\Psi}_k = \boldsymbol{\Psi}_k \boldsymbol{\Lambda}_k, \tag{20}$$

where the column vectors of $\boldsymbol{\Psi}_k \in \mathbb{R}^{N_k \times N_k}$ form an orthogonal basis of $C^k(K, \mathbb{R})$, satisfying

$$\boldsymbol{\Psi}_k^\top *_k \boldsymbol{\Psi}_k = \mathbf{I}_{N_k},$$

and $\boldsymbol{\Lambda}_k$ is the diagonal eigenvalue matrix. Eigenvectors with zero eigenvalues span $\ker \mathbf{L}_k$, whose dimension equals the $k$-th Betti number $b_k$, characterizing non-trivial topological structures such as non-contractible loops and cavities. Smaller non-zero eigenvalues correspond to large-scale modes with slow spatial variation, while larger eigenvalues correspond to localized, high-frequency modes.

Given any $\boldsymbol{\omega}_k \in C^k(K, \mathbb{R})$, its expansion under the Hodge spectral basis is

$$\boldsymbol{\omega}_k = \sum_{i=1}^{N_k} c_{k,i}\psi_{k,i}, \qquad c_{k,i} = \psi_{k,i}^\top *_k \boldsymbol{\omega}_k.$$

Here, $i \leq b_k$ corresponds to harmonic modes, and $i > b_k$ corresponds to non-harmonic modes. Truncating to the first $m_k$ eigenvectors yields the spectral subspace

$$\mathcal{V}_{\text{base}}^k = \operatorname{span}\{\psi_{k,1}, \ldots, \psi_{k,m_k}\},$$

which simultaneously contains all harmonic modes and several low-frequency non-harmonic modes, used to preserve topological information and approximate global large-scale behavior in a finite-dimensional space. The error from spectral truncation is mainly concentrated in high-frequency parts, which can be corrected by local operators in the fiber branch.

### B.4. Riemannian Geometry and Tangent Bundle

Physical operators at the continuous level typically depend on local metrics, curvature, and material property tensors, which are naturally defined on the tangent bundle. The tangent bundle structure provides the foundation for constructing local operators consistent with geometry in local Euclidean coordinates.

For any point $p \in \mathcal{M}$ on the manifold, the tangent space $T_p\mathcal{M}$ is a vector space isomorphic to $\mathbb{R}^n$, whose elements are tangent vectors at $p$. The Riemannian metric $g$ gives an inner product on $T_p\mathcal{M}$

$$\langle v, w \rangle_{g(p)} = g_p(v, w), \qquad v, w \in T_p\mathcal{M},$$

and the corresponding volume element $\mathrm{dvol}_g$. The collection of tangent spaces at all points forms the tangent bundle

$$T\mathcal{M} = \bigsqcup_{p \in \mathcal{M}} T_p\mathcal{M},$$

which is a differentiable manifold of dimension $2n$.

Through local coordinate charts $(U_\alpha, \varphi_\alpha)$, $\varphi_\alpha : U_\alpha \to \mathbb{R}^n$ introduces coordinates $x = (x^1, \ldots, x^n)$ within $U_\alpha$, with the corresponding coordinate basis

$$\left\{ \frac{\partial}{\partial x^1}, \ldots, \frac{\partial}{\partial x^n} \right\}.$$

In this basis, the metric tensor is represented as a symmetric positive definite matrix field $g_{ij}(x)$, i.e.,

$$g_p = \sum_{i,j=1}^{n} g_{ij}(x)\, \mathrm{d}x^i \otimes \mathrm{d}x^j.$$

Anisotropic diffusion tensors, stress tensors, and other quantities related to metric and material properties can be given in matrix form in local coordinates.

Many first-order and second-order partial differential operators have standard expressions in local coordinates of the tangent space. Taking the scalar field $u$ as an example, the anisotropic diffusion operator in local coordinates $x$ can be written as

$$\nabla \cdot \big( D(x) \nabla u(x) \big) = \frac{1}{\sqrt{|g(x)|}} \sum_{i,j=1}^{n} \frac{\partial}{\partial x^i} \left( \sqrt{|g(x)|}\, D^{ij}(x) \frac{\partial u}{\partial x^j} \right),$$

where $D(x)$ is a symmetric positive definite matrix field related to metric and material properties, and $|g(x)|$ is the determinant of the metric matrix. The advection operator can be written as

$$v(x) \cdot \nabla u(x) = \sum_{i=1}^{n} v^i(x) \frac{\partial u}{\partial x^i},$$

where $v(x)$ is a tangent vector field. For vector fields or higher-order form fields, the corresponding operators can be represented through combinations of covariant derivatives, exterior derivatives, and codifferentials, with coefficients also depending on local metrics and material properties.

The topological properties of exterior derivative $d$ and codifferential $\delta$ are independent of the metric, while the above diffusion and advection operators are highly sensitive to $g$ and material property tensors. In the Hodge Spectral Duality framework, the base space branch encodes the topology-dominated parts determined by $d$ and $\delta$ through the discrete Hodge Laplacian and its spectral structure, while the fiber branch models metric-dominated local effects through local coordinate representations on the tangent bundle and frequency domain operators, with consistency maintained through differentiable de Rham maps and orthogonal projections.

## C. Spectral Operator Theory and Subspace Derivation

This appendix formalizes the construction of truncated spectral subspaces, orthogonal projections, spectral derivative matrices, and harmonic projection matrices, based on the discrete exterior calculus and Hodge–de Rham spectral structure given in Appendix B, to support the simplified exposition in Sections 3.1 and 3.2 of the main text.

## C.1. Truncated Hodge Spectral Basis and Subspace Decomposition

From Appendix B, the $k$-th order discrete Hodge Laplacian $\mathbf{L}_k$ is a self-adjoint operator under the Hodge inner product

$$\langle \boldsymbol{\alpha}, \boldsymbol{\beta} \rangle_{*_k} = \boldsymbol{\alpha}^\top *_k \boldsymbol{\beta}, \qquad \boldsymbol{\alpha}, \boldsymbol{\beta} \in C^k(K, \mathbb{R})$$

Its spectral decomposition satisfies equation (2), and the eigenvector matrix $\boldsymbol{\Psi}_k$ can be chosen as a Hodge orthonormal basis, i.e., $\boldsymbol{\Psi}_k^\top *_k \boldsymbol{\Psi}_k = \mathbf{I}_{N_k}$. When $\dim \ker \mathbf{L}_k > 1$, different Hodge-orthonormal bases represent the same harmonic subspace $\ker \mathbf{L}_k$; cohomology classes remain invariant under coordinate rotations inside this subspace. Denote the corresponding eigenvalues as

$$0 = \lambda_{k,1} = \cdots = \lambda_{k,b_k} < \lambda_{k,b_k+1} \leq \cdots \leq \lambda_{k,N_k},$$

where $b_k = \dim \ker \mathbf{L}_k$ is the $k$-th Betti number.

To obtain a low-dimensional spectral representation for operator learning, truncate to the first $m_k$ eigenvectors

$$\boldsymbol{\Phi}_k = \left[ \psi_{k,1}, \ldots, \psi_{k,m_k} \right] \in \mathbb{R}^{N_k \times m_k}, \qquad \mathbf{L}_k \psi_{k,i} = \lambda_{k,i} \psi_{k,i}, \tag{21}$$

and perform a one-time orthogonalization under the Hodge inner product, so that

$$\boldsymbol{\Phi}_k^\top *_k \boldsymbol{\Phi}_k = \mathbf{I}_{m_k}. \tag{22}$$

This yields the truncated spectral subspace

$$\mathcal{V}_{\text{base}}^k = \text{span}(\boldsymbol{\Phi}_k) \subset C^k(K, \mathbb{R}), \tag{23}$$

which necessarily contains all harmonic modes as well as several lowest-frequency non-harmonic modes. The orthogonal complement under the Hodge inner product is defined as

$$\mathcal{V}_{\text{fiber}}^k = \left( \mathcal{V}_{\text{base}}^k \right)^{\perp_{*_k}} = \left\{ \boldsymbol{\eta} \in C^k(K, \mathbb{R}) \,\middle|\, \langle \boldsymbol{\eta}, \boldsymbol{\xi} \rangle_{*_k} = 0, \ \forall \boldsymbol{\xi} \in \mathcal{V}_{\text{base}}^k \right\}, \tag{24}$$

thus obtaining the approximate orthogonal decomposition

$$C^k(K, \mathbb{R}) = \mathcal{V}_{\text{base}}^k \oplus \mathcal{V}_{\text{fiber}}^k. \tag{25}$$

Based on equation (22), the Hodge orthogonal projection operator onto $\mathcal{V}_{\text{base}}^k$ is

$$\Pi_{\text{base}}^k = \boldsymbol{\Phi}_k \boldsymbol{\Phi}_k^\top *_k, \qquad \Pi_{\text{fiber}}^k = \mathbf{I} - \Pi_{\text{base}}^k, \tag{26}$$

where $\Pi_{\text{fiber}}^k$ is the projection onto the fiber subspace $\mathcal{V}_{\text{fiber}}^k$. Any field $\boldsymbol{\omega}_k \in C^k(K, \mathbb{R})$ can thus be uniquely decomposed as

$$\boldsymbol{\omega}_k = \boldsymbol{\omega}_{k,\text{base}} + \boldsymbol{\omega}_{k,\text{fiber}}, \qquad \boldsymbol{\omega}_{k,\text{base}} = \Pi_{\text{base}}^k \boldsymbol{\omega}_k, \quad \boldsymbol{\omega}_{k,\text{fiber}} = \Pi_{\text{fiber}}^k \boldsymbol{\omega}_k. \tag{27}$$

Compared to the exact discrete Hodge–de Rham decomposition, equations (25)–(27) concentrate harmonic and low-frequency modes into a finite-dimensional subspace through spectral truncation, while concentrating the remaining high-frequency degrees of freedom into the orthogonal complement.

## C.2. Splitting of Topological and Geometric Operators

At the continuous level, the control operator $\mathcal{A}^k$ can be abstractly decomposed into a topology-dominated part generated by exterior derivatives and codifferentials, and a geometry-dominated part dependent on metric and material property tensors

$$\mathcal{A}^k = \mathcal{A}_{\text{Topo}}^k + \mathcal{A}_{\text{Geom}}^k, \tag{28}$$

where $\mathcal{A}_{\text{Topo}}^k$ is generated by $d$, $\delta$, and the Hodge–de Rham Laplacian, mainly characterizing cohomological constraints and conservation structures; $\mathcal{A}_{\text{Geom}}^k$ contains diffusion, advection, and source/sink terms dependent on $g$ and material property tensors. The corresponding solution operator can be formally written as

$$\mathcal{G}^k \approx \mathcal{G}_{\text{base}}^k + \mathcal{G}_{\text{fiber}}^k, \qquad \mathcal{G}_{\text{base}}^k : C^k(K, \mathbb{R}) \to \mathcal{V}_{\text{base}}^k, \quad \mathcal{G}_{\text{fiber}}^k : C^k(K, \mathbb{R}) \to \mathcal{V}_{\text{fiber}}^k. \tag{29}$$

The neural operator branches $\mathcal{G}_{\text{base},\theta}^k$ and $\mathcal{G}_{\text{fiber},\theta}^k$ in the main text equation (3) are precisely parameterized approximations of equation (29).

## C.3. Spectral Derivative Matrices and Harmonic Projection

To preserve the discrete exterior calculus structure in the truncated spectral domain, the discrete exterior derivative $d_k$ and discrete codifferential $\delta_k$ from DEC are projected onto the spectral basis $\mathbf{\Phi}_k$, yielding spectral derivative matrices

$$\mathcal{M}_d^{(k)} = \mathbf{\Phi}_{k+1}^\top *_{k+1} d_k \mathbf{\Phi}_k, \qquad \mathcal{M}_\delta^{(k)} = \mathbf{\Phi}_{k-1}^\top *_{k-1} \delta_k \mathbf{\Phi}_k. \tag{30}$$

For any field $\boldsymbol{\omega}_k$, its spectral coefficients are $\mathbf{c}_k = \mathbf{\Phi}_k^\top *_k \boldsymbol{\omega}_k$. In the truncated subspace, the spectral representations of the discrete exterior derivative and codifferential approximately satisfy

$$\mathbf{\Phi}_{k+1}^\top *_{k+1} d_k \boldsymbol{\omega}_k \approx \mathcal{M}_d^{(k)} \mathbf{c}_k, \qquad \mathbf{\Phi}_{k-1}^\top *_{k-1} \delta_k \boldsymbol{\omega}_k \approx \mathcal{M}_\delta^{(k)} \mathbf{c}_k,$$

so $\mathcal{M}_d^{(k)}$ and $\mathcal{M}_\delta^{(k)}$ preserve the algebraic structure of the original differential complex in the spectral domain. The spectral features in Section 3.2 of the main text can be written as

$$\mathbf{q}_k^{(\ell)} = \operatorname{concat}\big(\mathbf{c}_k^{(\ell)}, \mathcal{M}_d^{(k)} \mathbf{c}_k^{(\ell)}, \mathcal{M}_\delta^{(k)} \mathbf{c}_k^{(\ell)}\big). \tag{31}$$

This combined feature is then fed into $\operatorname{gMLP}_k$ defined in the main text equation (5), using the multiplicative structure of the gating mechanism to approximate nonlinear mode coupling between physical fields.

The harmonic projection is based on the zero eigenvalue structure of $\mathbf{L}_k$. Denote

$$\mathcal{I}_H^k = \big\{i \in \{1, \ldots, m_k\} \,\big|\, \lambda_{k,i} = 0\big\}$$

as the index set of harmonic modes in the truncated spectral basis, and define the diagonal projection matrix $\mathbf{P}_H^k \in \mathbb{R}^{m_k \times m_k}$ as

$$\big[\mathbf{P}_H^k\big]_{ij} = \begin{cases} 1, & i = j \in \mathcal{I}_H^k, \\ 0, & \text{otherwise}, \end{cases}$$

then applying the harmonic hard constraint to the layer $\ell$ spectral update result $\tilde{\mathbf{c}}_k^{(\ell)}$ can be written as

$$\tilde{\mathbf{c}}_k^{(\ell)} \leftarrow \tilde{\mathbf{c}}_k^{(\ell)} + \mathbf{P}_H^k \big(\mathbf{c}_k^{(\ell)} - \tilde{\mathbf{c}}_k^{(\ell)}\big), \tag{32}$$

which is equivalent to forcibly restoring the components at harmonic indices to the input coefficients $\mathbf{c}_k^{(\ell)}$, thereby exactly preserving the corresponding cohomology classes and global conserved quantities at each layer. In time-dependent settings, these coefficients are determined by the reference state supplied by the problem data, and source-driven harmonic changes can be incorporated through task-specific coefficient updates.

Finally, the base space reconstruction formula is written as

$$\boldsymbol{\omega}_{k,\text{base}}^{(\ell+1)} = \mathbf{\Phi}_k \tilde{\mathbf{c}}_k^{(\ell)}, \tag{33}$$

which together with equation (4) forms a closed loop of spectral projection and reconstruction, ensuring that the base space branch output always lies in $\mathcal{V}_{\text{base}}^k$.

# D. Consistency, Stability, and Resolution Requirements of Ambient Fiber Embedding

This section analyzes the consistency of the composite operator $\mathcal{R} \circ \mathcal{A}_{\text{amb}} \circ \iota_{h,\epsilon}$, consisting of discrete differential form lifting, spectral operator convolution, and pullback operations, relative to the manifold-intrinsic operator $\mathcal{A}_\mathcal{M}$. Addressing the derivative discontinuity of low-order Whitney forms at element interfaces and the resolution of boundary layer features on anisotropic meshes, this section provides error bound analysis based on Reach conditions (Federer, 1959), Nyquist sampling, and anisotropic kernels, and clarifies the adaptive adjustment mechanisms for model hyperparameters.

## D.1. Geometric Assumptions and Discretization Setup

To ensure rigor in the analysis, we first clarify the basic assumptions on geometry and discretization. Assume the manifold $\mathcal{M}$ has Reach $\tau_\mathcal{M} > 0$, and the simplicial complex $K$ approximates $\mathcal{M}$ with Hausdorff error $O(h)$. For mesh quality, define

the condition number $\gamma_K$ of the geometric matrix $G_K$ for each element, and let the global maximum condition number $\gamma := \sup_K \text{cond}(G_K) < \infty$. This parameter $\gamma$ characterizes the degree of mesh anisotropy, allowing elements with extreme aspect ratios, but the relevant error constants will increase with $\gamma$.

In the ambient embedding process, the mollification bandwidth $\epsilon$ must satisfy $0 < \epsilon < c\tau_{\mathcal{M}}$, where $c \in (0, 1)$, to ensure uniqueness of the nearest point projection and prevent non-physical topological shortcuts. For numerical implementation, the ambient voxel grid step size $h_{\text{aux}}$ constitutes a hard constraint on resolution. According to the Nyquist sampling theorem, $h_{\text{aux}}$ is required to satisfy $h_{\text{aux}} \leq \min(\epsilon/2, c_0 \delta_{\text{res}}/4)$, where $\delta_{\text{res}}$ is the smallest characteristic scale that needs to be resolved in the physical problem. This condition indicates that the ambient grid must have sufficient resolution to support the chosen mollification scale and physical features.

To mitigate aliasing problems caused by anisotropic meshes, theoretically one should adopt metric-adaptive anisotropic kernel functions, with separate settings for normal bandwidth $\epsilon_n$ and tangential bandwidth $\epsilon_t$, satisfying $\epsilon_n \leq \min(\tau_{\mathcal{M}}/2, \delta_{\text{res}}/3)$ and $\epsilon_t \asymp \alpha h_{\text{loc}} \gamma^{1/2}$. This setting ensures that high-frequency features in the normal direction are not over-smoothed while maintaining sufficient coverage in the tangential direction to suppress interpolation noise.

### D.2. Stability and Consistency of Whitney Extension

Let $W_h$ denote the piecewise $H^1$ field obtained by Whitney reconstruction from discrete forms in $C^k(K)$. Since low-order Whitney forms have derivative jumps across element interfaces, directly applying differential operators to them introduces Dirac-type singularities. The ambient mollification operator $\iota_{h,\epsilon}$ serves a regularization role here, transforming $W_h$ into the smooth ambient field $u_\epsilon$.

Regarding the stability of this process, it can be shown that the gradient norm after mollification is controlled by the original discrete field, i.e., $\|\nabla u_\epsilon\|_{L^2(\mathbb{R}^d)} \leq C(\gamma)\|\nabla W_h\|_{L^2(\mathcal{M})}$, where the constant $C(\gamma)$ increases monotonically with mesh anisotropy. Regarding approximation consistency, for target fields $u$ with bounded curvature and second-order regularity, the error restricted to the manifold satisfies

$$\|\nabla(u_\epsilon|_{\mathcal{M}}) - \nabla u\|_{L^2(\mathcal{M})} \leq C(\gamma)\left(\epsilon + \frac{h}{\epsilon}\right). \tag{34}$$

Equation (34) reveals the trade-off mechanism for mollification bandwidth $\epsilon$. The first term $\epsilon$ comes from bias introduced by mollification itself, while the second term $h/\epsilon$ reflects the residual variance from mollification failing to completely eliminate discrete mesh roughness and derivative jumps. Therefore, the theoretically optimal bandwidth should be chosen at the $\epsilon \asymp h^{1/2}$ level.

In engineering implementation, the mollification bandwidth $\epsilon$ is not a completely independent hyperparameter, but is implicitly locked by the ambient grid resolution $h_{\text{aux}}$ and the effective support radius of the interpolation kernel. From resolution constraints, the lower bound of the effective bandwidth is limited by voxel size, i.e., $\epsilon_{\text{eff}} \geq h_{\text{aux}}$.

### D.3. Total Error Decomposition and Adaptive Correction

Combining the above geometric embedding error and spectral operator approximation error, the overall consistency of the Fiber branch can be described by total error decomposition. Let $\mathcal{A}_{\text{amb}}$ be the ambient domain FNO operator, $\mathcal{R}$ be the pullback operator, and assume the output is corrected by orthogonal projection $(\mathbf{I} - \Pi_{\text{base}}^k)$. Under the aforementioned geometric assumptions, the total error $E_{\text{total}}$ has an upper bound

$$E_{\text{total}} := \|(\mathbf{I} - \Pi_{\text{base}}^k)(\mathcal{R}\mathcal{A}_{\text{amb}}\iota_{h,\epsilon} - \mathcal{A}_{\mathcal{M}})\| \leq C\left[E_{\text{geom}}(\epsilon, h, \gamma) + E_{\text{vox}}(h_{\text{aux}}) + E_{\text{spec}}(\xi_{\text{task}})\right]. \tag{35}$$

Here, $E_{\text{geom}} \asymp \epsilon + h/\epsilon$ is the geometric discretization and mollification error, $E_{\text{vox}} \asymp h_{\text{aux}}$ is the voxelization discretization error, and $E_{\text{spec}}$ is the frequency domain truncation error.

For cases where mesh resolution may be insufficient or aspect ratios are extremely poor, this framework achieves implicit adaptive adjustment through machine learning mechanisms. First, the learnable scalar $\lambda$ in the model serves as a confidence gate. If the ambient grid resolution is insufficient to resolve high-frequency geometric features in certain regions, causing the Fiber branch output to be noisy or have large deviations, the optimization process will drive $\lambda$ to decay. This allows the model to automatically degrade to a topology-preserving solution dominated by the Base branch in extreme cases, thereby ensuring numerical stability. Second, the spectral kernel weights $\mathbf{R}_{\text{loc}}$ of the ambient FNO will automatically adapt to

the frequency domain truncation introduced by mollification during training. For aliased components beyond the Nyquist frequency $\pi/h_{\mathrm{aux}}$, the network tends to learn near-zero gains, thus acting as a data-driven low-pass filter.

Finally, the orthogonal projection operator $(\mathbf{I} - \Pi_{\mathrm{base}}^k)$ constitutes an algebraic-level safety barrier. Even if the ambient branch introduces erroneous low-frequency modes or conservation-violating artifacts due to resolution limitations, these components will be eliminated by the projection operation.

The hard constraint on the harmonic subspace acts as a physically consistent regularization mechanism. Since measurement noise typically violates conservation laws and falls in the $\operatorname{im} d$ or $\operatorname{im} \delta$ spaces orthogonal to the harmonic kernel, enforcing topological constraints removes these non-physical residuals during loss minimization and projects noisy data onto the conservative solution manifold.

Additionally, for high-frequency discretization artifacts introduced by the voxelization process, the finite spectral bandwidth property of the ambient FNO acts as an intrinsic low-pass filter, effectively attenuating mesh-scale aliased components remaining after orthogonal projection.

### D.4. Efficiency and Robustness in $\mathbb{R}^3$ Embeddings

This method primarily targets physical objects embedded in $\mathbb{R}^3$, such as aerodynamic shapes and biological tissues. In these scenarios, the computational efficiency gains from Ambient FNO significantly outweigh the accuracy loss introduced by the interpolation process. The lift operator $\iota$ has a built-in controlled convolution kernel that mathematically guarantees the signal satisfies band-limited conditions before entering the ambient grid, thereby avoiding severe aliasing. Even if the ambient branch produces high-frequency noise, the projection operator $\mathbf{I} - \Pi_{\mathrm{base}}$ ensures this noise is confined to the orthogonal complement space and absolutely cannot leak into the topology-dominated low-frequency subspace. This mechanism guarantees the immunity of physical conservation laws to numerical artifacts, while the remaining small high-frequency residuals are naturally smoothed through the spectral bias property of neural networks.

The splatting mapping from discrete simplices to the background grid is essentially kernel density estimation. Regardless of how the manifold geometry curls or self-occludes, this process only involves local additive operations on simplices. Its computational complexity is $O(N_{\mathrm{simplex}} \times K^3)$, where $K$ is the kernel width. This complexity is only linear with respect to the number of elements and completely independent of manifold curvature or topological complexity. Physically, physical fields on adjacent patches appear as a superposition or transition in ambient space, and splatting automatically handles this property without additional geometric detection costs. For extremely irregular geometries, sparse octree or hash grid techniques can concentrate computation in the manifold neighborhood and scale with manifold surface area.

### D.5. Boundary Regularity and Suppression of Gibbs Artifacts

Addressing the binary mask step at the manifold boundary $\partial\mathcal{M}$ and periodic splicing discontinuities at computational box boundaries, this section analyzes the Gibbs ringing phenomenon caused thereby and its suppression strategies. If the Fourier transform is directly applied to the binarized embedded field, spatial discontinuities will cause high-frequency coefficient decay rates to degrade from $O(|k|^{-p})$ to $O(|k|^{-1})$, producing significant spatial domain ringing that may confuse physical high-frequency features. To address this, we introduce a processing mechanism combining geometric regularization and operator correction.

First, at the geometric level, $C^r$ continuous transition regions are constructed to replace binary steps. Based on the manifold signed distance field $d(x)$, construct a smooth soft mask $m_{\epsilon_n}(x)$, where $\epsilon_n$ is the normal bandwidth, satisfying $\epsilon_n \lesssim \min(\tau_{\mathcal{M}}/2, \delta_{\mathrm{res}}/3)$. Define the extended field $u_{\mathrm{ext}}(x) = m_{\epsilon_n}(x)\iota(u)(x) + (1 - m_{\epsilon_n}(x))E[u](x)$, where $E[u]$ is a boundary-consistent extension operator ensuring $u_{\mathrm{ext}}$ satisfies at least $C^1$ continuity at boundaries. This regularization treatment restores the Fourier coefficient decay rate to $O(|k|^{-(r+1)})$, fundamentally weakening the energy source of the Gibbs phenomenon. Meanwhile, to handle non-periodicity at computational box boundaries, smooth window functions $w(x)$ or absorbing layers are applied at FFT domain edges to eliminate boundary artifacts introduced by periodic wrapping.

Second, at the algebraic and operator level, orthogonal projection $(\mathbf{I} - \Pi_{\mathrm{base}}^k)$ ensures that any artifacts falling into harmonic or low-frequency subspaces due to improper boundary handling are removed, ensuring ringing noise does not pollute global topological invariants and conservation laws. The remaining high-frequency ringing is confined to the Fiber subspace and suppressed through introducing boundary band energy regularization terms $\mathcal{L}_{\mathrm{ring}} = \lambda \int |\nabla m_{\epsilon_n}|^2 |\omega_{\mathrm{fiber}}|^2 dx$ in the loss function as well as frequency domain high-frequency penalties.

Under the above treatments, the total approximation error can be further decomposed into a form including boundary regularity

$$E_{\text{tot}} \leq C \left[ E_{\text{geom}}(\epsilon_n, h) + E_{\text{vox}}(h_{\text{aux}}) + E_{\text{cut}}(r, \xi_{\text{cut}}) + E_{\text{wrap}}(L_{\text{pml}}) \right]. \tag{36}$$

Here $E_{\text{cut}}$ decays significantly as the transition region regularity $r$ increases, and $E_{\text{wrap}}$ decreases as the absorbing layer thickness $L_{\text{pml}}$ increases. This error bound indicates that by improving the regularity of the embedded field through soft masks and boundary processing, Gibbs error can be effectively controlled, validating the numerical effectiveness of this framework on bounded manifolds.

## E. Theoretical Justification of the Commutator Corrector

This section provides the theoretical justification for the commutator correction term $\mathcal{C}_\theta^{(\ell)}$. We first derive the analytical form of the operator commutator $[\mathcal{A}_{\text{Topo}}^k, \mathcal{A}_{\text{Geom}}^k]$, then prove that within the Hodge Spectral Duality framework, the interaction features $\mathbf{z}^{(\ell)}$ constitute the complete derivative proxy required by this commutator.

### E.1. Analytical Form of the Commutator

Consider $k$-th order differential form fields $\omega$ defined on a Riemannian manifold $(\mathcal{M}, g)$ with boundary. The topology-dominated operator $\mathcal{A}_{\text{Topo}}^k$ is generated by exterior derivative complex operators $(d, \delta, \Delta_k)$ and typically does not explicitly depend on local metrics; while the geometry-dominated operator $\mathcal{A}_{\text{Geom}}^k$ (such as anisotropic diffusion, advection) explicitly depends on position-dependent material property tensors $\kappa(x)$ and Riemannian metric $g(x)$.

Using the Leibniz rule on Riemannian manifolds, the commutator of the two is typically non-zero. Taking scalar fields as an example, let $\mathcal{A}_{\text{Topo}} = \Delta$ (Laplace–Beltrami operator), $\mathcal{A}_{\text{Geom}} = M_\kappa$ (multiplication operator by $\kappa(x)$), then:

$$[\Delta, M_\kappa]u = \Delta(\kappa u) - \kappa \Delta u = (\Delta \kappa)u + 2 \langle \nabla \kappa, \nabla u \rangle_g. \tag{37}$$

The above equation shows that the residual term is driven by two parts: (1) higher-order derivatives of geometric parameters $(\Delta \kappa)$; (2) coupling between geometric parameter gradients and physical field gradients $(\nabla \kappa \cdot \nabla u)$.

Generalizing to $k$-th order forms, the general form of the commutator can be written as a local differential operator $\mathcal{F}_{\text{comm}}$:

$$[\mathcal{A}_{\text{Topo}}^k, \mathcal{A}_{\text{Geom}}^k]\, \omega \;=\; \mathcal{F}_{\text{comm}}\big(\omega,\, d\omega,\, \delta\omega,\, \kappa,\, d\kappa,\, g,\, \nabla g\big). \tag{38}$$

This depends on zeroth-order field values $\omega$ and $\kappa$ together with their first-order derivative information. Therefore, any network $\mathcal{C}_\theta$ attempting to correct this spectral-geometric splitting error must explicitly or implicitly access this set of derivative information.

### E.2. Spectral Encoding as Derivative Proxy for Fields

To capture the dependence on $d\omega$ and $\delta\omega$ in equation (38), we utilize the spectral structure of the Base branch. Recalling the spectral coefficients $\mathbf{c}_k$ and spectral derivative matrices $\mathcal{M}_d^{(k)}, \mathcal{M}_\delta^{(k)}$ defined in Section 3.2 of the main text, we have the following algebraic identities:

$$d_k \boldsymbol{\omega}_k \approx \boldsymbol{\Phi}_{k+1}\big(\mathcal{M}_d^{(k)} \mathbf{c}_k\big), \qquad \delta_k \boldsymbol{\omega}_k \approx \boldsymbol{\Phi}_{k-1}\big(\mathcal{M}_\delta^{(k)} \mathbf{c}_k\big). \tag{39}$$

This means that within the truncated spectral subspace $\mathcal{V}_{\text{base}}^k$, there exists a linear isomorphism between the vector group $(\mathbf{c}_k, \mathcal{M}_d^{(k)} \mathbf{c}_k, \mathcal{M}_\delta^{(k)} \mathbf{c}_k)$ and the physical field and its first-order derivatives $(\omega, d\omega, \delta\omega)$.

The interaction features $\mathbf{z}^{(\ell)}$ we construct in Section 3.4 explicitly concatenate this set of vectors. Therefore, for a fixed simplicial complex, $\mathbf{z}^{(\ell)}$ completely encodes the first-order differential structure of the field in the discrete sense, enabling the MLP to theoretically recover $d\omega$ and $\delta\omega$ from input features.

### E.3. Geometric Interaction in the Fiber Branch

To capture the dependence on geometric derivatives $\nabla \kappa, \nabla g$ in equation (38), we rely on the convolutional properties of the Fiber branch. The Fiber branch embeds discrete forms into the auxiliary Euclidean grid through the lift mapping $\iota$ and

applies frequency domain convolution operators $\mathbf{R}_{\text{loc}}^{(\ell)}$. Convolution operations are locally equivalent to weighted differences, so the output features $\mathbf{u}_{\text{geom}}^{(\ell)}$ implicitly contain the spatial variation rates (i.e., derivative information) of input parameters.

In summary, the two parts of the interaction features $\mathbf{z}^{(\ell)}$ respectively provide: **Base part** explicitly provides algebraic derivatives of physical fields $(d\omega, \delta\omega)$; **Fiber part** implicitly provides numerical derivatives of geometric and material property parameters $(\nabla\kappa, \nabla g)$.

Based on the Universal Approximation Theorem, the parameterized MLP $\mathcal{C}_\theta^{(\ell)}$ can utilize this complete local information to approximate the nonlinear commutator residual $\mathcal{F}_{\text{comm}}$. Finally, through orthogonal projection $(\mathbf{I} - \Pi_{\text{base}}^k)$, we restrict this correction to the orthogonal complement space, thereby ensuring that the underlying topological conservation laws are not corrupted by approximation errors.

# F. Computational Complexity Analysis

## F.1. Computational Complexity

The Hodge Spectral Duality framework adopts an offline-online decoupling strategy, concentrating geometry-dependent spectral operations into a one-time preprocessing stage while achieving approximately linear online inference complexity with respect to mesh scale.

**Offline Stage: DEC Assembly and Spectral Decomposition.**   The offline stage constructs DEC operators on the simplicial complex $K$, including oriented boundary matrices $\mathbf{B}_k$, discrete exterior derivatives $d_k$, discrete Hodge stars $*_k$, discrete codifferentials $\delta_k$, and the discrete Hodge Laplacian $\mathbf{L}_k$. These operators are highly sparse: each $k$-simplex is adjacent to a bounded number of $(k \pm 1)$-simplices, yielding $\text{nnz}(\mathbf{L}_k) = \mathcal{O}(N_k)$. Assembly involves only sparse matrix multiplications with favorable memory access patterns, achieving complexity $\mathcal{O}(N_k)$.

For spectral decomposition, we employ the Shift-Invert Spectral Transformation for low-frequency eigenpairs. We solve for the largest eigenvalues of the transformed operator $(\mathbf{L}_k - \sigma\mathbf{I})^{-1}$ with shift $\sigma \approx 0$, which magnifies the spectral gap between target low-frequency modes and the remainder of the spectrum and accelerates Krylov subspace convergence. Combined with sparse direct factorization of the shifted operator, the complexity for extracting $m_k$ eigenpairs becomes $\mathcal{O}(m_k \, \text{nnz}(\mathbf{L}_k)) \approx \mathcal{O}(m_k N_k)$, where the iteration count implicit in $m_k$ remains small due to rapid convergence. Precomputing spectral derivative matrices $\mathcal{M}_d^{(k)}, \mathcal{M}_\delta^{(k)}$ and projection operators $\Pi_{\text{base}}^k$ from $\mathbf{\Phi}_k$ and sparse operators $d_k, \delta_k$ also achieves $\mathcal{O}(m_k N_k)$ complexity.

**Online Stage: Base Space and Fiber Branches.**   For the base space branch, spectral projection $\mathbf{c}_k^{(\ell)} = \mathbf{\Phi}_k^\top *_k \boldsymbol{\omega}_k^{(\ell)}$ and reconstruction $\tilde{\boldsymbol{\omega}}_k^{(\ell)} = \mathbf{\Phi}_k \tilde{\mathbf{c}}_k^{(\ell)}$ are dense matrix-vector multiplications with complexity $\mathcal{O}(N_k m_k)$. Since $m_k \ll N_k$ (typically $m_k \sim 64$), these operations are implemented as highly parallel GEMM kernels with arithmetic intensity well-suited for GPU acceleration. Furthermore, the exact sequence property $d \circ d = 0$ and $\delta \circ \delta = 0$ is preserved at the spectral level, allowing certain higher-order derivative compositions to be short-circuited to zero without floating-point operations.

For the Fiber branch, point-voxel mapping $\iota$ interpolates signals onto a regular background grid with resolution $R$ and total voxels $V = R^3$. This mapping and its inverse involve sparse interpolation with complexity $\mathcal{O}(N_k)$. The 3D FNO executed on the background grid has complexity $\mathcal{O}(V \log V)$, where $V$ is a fixed constant independent of mesh scale. This replaces the $\mathcal{O}(N_k)$ irregular memory accesses required for per-vertex tangent space convolutions with structured FFT operations on regular grids, eliminating sparse index lookup overhead.

The total complexity of a single forward pass is $\mathcal{O}(N_k m_k) + \mathcal{O}(N_k + V \log V) \approx \mathcal{O}(N_k)$, achieving linear scaling with mesh resolution. Compared to higher-order graph neural networks based on sparse message passing, this framework improves hardware utilization through low-dimensional spectral projection and structured FFT operations. Compared to intrinsic geometric deep learning methods, ambient space approximation decouples high-frequency geometric processing complexity from mesh resolution.

## F.2. Scalability and Precomputation Cost

The offline-online decomposition constitutes a computational arbitrage strategy. For physical simulation tasks, geometric meshes are typically fixed or undergo isometric deformations, making the one-time preprocessing investment negligible

relative to thousands of training iterations with $\mathcal{O}(N_k)$ online complexity.

The Shift-Invert strategy completes spectral decomposition within minutes for million-scale meshes ($N \sim 10^6$). Sparse direct factorization of $(\mathbf{L}_k - \sigma\mathbf{I})$ exploits the bounded fill-in characteristic of discretizations on manifolds, and the rapid Krylov convergence induced by spectral gap magnification keeps iteration counts low regardless of mesh scale.

For dynamic scenarios requiring frequent mesh topology updates, approximate spectral solvers based on multilevel algorithms or hierarchical matrix techniques can further reduce preprocessing overhead. Such extensions are beyond the scope of this paper, which focuses on establishing the foundational architecture.

**Resolution Efficiency.** A fundamental advantage of the dual-branch architecture lies in its decoupling of physical fidelity from geometric sampling density. Regardless of mesh resolution, the base space branch processes only the first $m_k$ spectral coefficients (e.g., $m_k = 64$ or 128), confining global topological information to a fixed low-dimensional subspace whose computational cost is nearly independent of $N_k$. The Fiber branch handles high-frequency details through ambient space FFT with complexity $\mathcal{O}(V \log V)$ independent of mesh density, avoiding the global mesh refinement traditionally required to capture boundary layers or localized features.

As demonstrated in our resolution robustness analysis (Section 4.8), HSD maintains consistent accuracy even when inference mesh density is significantly reduced, and models trained on coarse meshes transfer zero-shot to high-resolution meshes (7000+ vertices) with only marginal error increase. This resolution efficiency provides empirical evidence that the prohibitive computational complexity often associated with manifold PDEs is not a fundamental physical necessity. Furthermore, aggressive mesh decimation can be applied as an additional computational reduction strategy: while subsampling degrades pointwise numerical precision in high-frequency components, the dominant low-frequency spectral modes that govern global physical behavior remain well-resolved.

# G. Training Details and Hyperparameter Configuration

This section provides detailed information on experimental training costs, computational environment, optimization strategies, hyperparameter settings, and training procedures for each task.

## G.1. Dataset Splitting Strategy

For all three tasks, we generated 3,000 simulation samples. The dataset was split using a strict temporal partitioning strategy to prevent data leakage: the test set was first separated (20% of total data), then the remaining data was divided into training and validation sets (validation set comprising 15% of the remaining data). Detailed split statistics are shown in Table 5.

*Table 5.* Dataset split statistics.

| Dataset | Proportion | Samples | Purpose |
|---|---|---|---|
| Training set | 68% | 2,040 | Gradient updates and parameter optimization |
| Validation set | 12% | 360 | Hyperparameter tuning and model selection (early stopping) |
| Test set | 20% | 600 | Final performance evaluation and metric computation |
| Total | 100% | 3,000 | — |

## G.2. Auxiliary Evaluations

*Stress tests.* Ellipsoid Aero evaluates boundary-driven external-flow reconstruction on genus-zero ellipsoidal surfaces with controlled aspect ratio, curvature variation, vortex-pair forcing, and global moment coupling. Torus Helmholtz evaluates a Helmholtz-type response on a genus-one torus, where two independent harmonic 1-form directions coexist with mid- and high-frequency oscillatory forcing. Table 6 reports the matched-parameter comparison with recent attention/operator baselines GNOT (Hao et al., 2023), ONO (Xiao et al., 2024), and HAMLET (Bryutkin et al., 2024), together with input-format robustness across mesh, point-cloud, and graph discretizations. For point-cloud inputs, the complex and Laplacian are reconstructed from neighborhood connectivity and heat-kernel weights before applying the same Hodge spectral pipeline.

---

**Algorithm 1** Forward Pass of Hodge Spectral Duality Operator

---

1: **Input:** Discrete right-hand side $\boldsymbol{f}_k$, spectral basis $\boldsymbol{\Phi}_k$, Hodge Laplacian $\mathbf{L}_k$, precomputed operators $d_k, \delta_k, *_k, \Pi_{\text{base}}^k, \mathcal{M}_d^{(k)}, \mathcal{M}_\delta^{(k)}$

2: **Output:** Approximate solution $\boldsymbol{\omega}_k^\star$

3: **Offline Precomputation (completed once when geometry is fixed)**

4: Construct boundary matrices $\mathbf{B}_k$, discrete exterior derivative $d_k$, Hodge star operator $*_k$, and Hodge Laplacian $\mathbf{L}_k$

5: Compute the first $m_k$ eigenpairs of $\mathbf{L}_k$ to obtain spectral basis $\boldsymbol{\Phi}_k$

6: Compute spectral derivative operators $\mathcal{M}_d^{(k)}, \mathcal{M}_\delta^{(k)}$ and projection operator $\Pi_{\text{base}}^k$

7: **Online Forward Pass**

8: Initialize $\boldsymbol{\omega}_k^{(0)}$ from the problem data, such as an initial condition, boundary-consistent pre-solve, or source-derived reference state.

9: **for** $\ell = 0$ to $L - 1$ **do**

10: $\quad \mathbf{c}_k^{(\ell)} \leftarrow \boldsymbol{\Phi}_k^\top *_k \boldsymbol{\omega}_k^{(\ell)}$ $\qquad\qquad\qquad\qquad\qquad\qquad$ *# Lift to Hodge spectral domain*

11: $\quad \mathbf{q}_k^{(\ell)} \leftarrow \text{concat}\big(\mathbf{c}_k^{(\ell)}, \mathcal{M}_d^{(k)}\mathbf{c}_k^{(\ell)}, \mathcal{M}_\delta^{(k)}\mathbf{c}_k^{(\ell)}\big)$ $\qquad\quad$ *# Spectral features w/ derivatives*

12: $\quad \tilde{\mathbf{c}}_k^{(\ell)} \leftarrow \text{MLP}_k\big(\mathbf{q}_k^{(\ell)}\big)$

13: $\quad$ Apply harmonic hard constraint on $\tilde{\mathbf{c}}_k^{(\ell)}$ at $\mathcal{I}_H^k$

14: $\quad \boldsymbol{\omega}_{k,\text{base}}^{(\ell+1)} \leftarrow \boldsymbol{\Phi}_k \tilde{\mathbf{c}}_k^{(\ell)}$ $\qquad\qquad\qquad\qquad\qquad$ *# Reconstruct base space component*

15: $\quad \mathbf{u}_k^{(\ell)} \leftarrow \iota\big(\boldsymbol{\omega}_k^{(\ell)}, \boldsymbol{\omega}_{k,\text{base}}^{(\ell+1)}, \boldsymbol{f}_k, \text{geometry}\big)$ $\qquad\qquad$ *# Map to ambient background grid*

16: $\quad \mathbf{u}_{\text{geom}}^{(\ell)} \leftarrow \mathcal{F}^{-1}\mathbf{R}_{\text{loc}}^{(\ell)}\mathcal{F}\big(\mathbf{u}_k^{(\ell)}\big)$ $\qquad\qquad\qquad$ *# Ambient space FNO convolution*

17: $\quad \tilde{\boldsymbol{\omega}}_{k,\text{geom}}^{(\ell)} \leftarrow \mathcal{R}\big(\mathbf{u}_{\text{geom}}^{(\ell)}\big)$ $\qquad\qquad\qquad\qquad\qquad$ *# Pull back to simplicial complex*

18: $\quad \mathbf{z}^{(\ell)} \leftarrow \iota\big(\boldsymbol{\omega}_k^{(\ell)}\big) \oplus \mathbf{q}_k^{(\ell)}$ $\qquad\qquad\qquad\qquad$ *# Coupling features (reuse derivatives)*

19: $\quad \boldsymbol{\omega}_{k,\text{int}}^{(\ell)} \leftarrow \mathcal{C}_\theta^{(\ell)}\big(\mathbf{z}^{(\ell)}\big)$ $\qquad\qquad\qquad\qquad\quad$ *# Nonlinear commutator correction*

20: $\quad \boldsymbol{\omega}_{k,\text{fiber}}^{(\ell+1)} \leftarrow \big(\mathbf{I} - \Pi_{\text{base}}^k\big)\big(\tilde{\boldsymbol{\omega}}_{k,\text{geom}}^{(\ell)} + \boldsymbol{\omega}_{k,\text{int}}^{(\ell)}\big)$ $\qquad\qquad$ *# Orthogonal projection*

21: $\quad \boldsymbol{\omega}_k^{(\ell+1)} \leftarrow \boldsymbol{\omega}_{k,\text{base}}^{(\ell+1)} + \boldsymbol{\omega}_{k,\text{fiber}}^{(\ell+1)}$

22: **end for**

23: **Return** $\boldsymbol{\omega}_k^\star \leftarrow \boldsymbol{\omega}_k^{(L)}$

---

Across mesh, point-cloud, and graph inputs, the maximum relative variation is below $8.5\%$ on Ellipsoid Aero and below $6.0\%$ on Torus Helmholtz.

*Table 6.* Auxiliary stress-test results. Relative $L^2$ errors are reported for both tasks.

| Model / Input | Ellipsoid Aero | Torus Helmholtz |
|---|---|---|
| *Recent attention/operator baselines* | | |
| HSD (Ours) | **0.037** | **0.058** |
| GNOT | 0.144 | 0.277 |
| ONO | 0.155 | 0.262 |
| HAMLET | 0.159 | 0.250 |
| DeepONet | 0.221 | 0.652 |
| Geo-FNO | 0.240 | 0.449 |
| FNO | 0.246 | 0.418 |
| *Input discretization for HSD* | | |
| Mesh | 0.036 | 0.054 |
| Point cloud | 0.039 | 0.057 |
| Graph | 0.036 | 0.056 |

*Topological identification.* This evaluation uses the algebraic relation $\ker \Delta_k = \ker d_k \cap \ker \delta_k$ and $b_k = \dim \ker \Delta_k$, which identifies Betti numbers with the dimension of the harmonic nullspace. We test whether Hodge-based spectral features retain enough global structure to predict these invariants from point-cloud inputs. The resulting classifier reached $91.0\%$ accuracy, compared with $84.6\%$ for a Euclidean-distance baseline.

## G.3. Detailed Computational Cost Analysis

Table 7 presents a detailed comparison of time and memory consumption for each model under identical hardware conditions (a single NVIDIA RTX PRO 6000 Blackwell Workstation Edition GPU). The total time for HSD includes one-time spectral basis precomputation overhead (indicated in parentheses).

*Table 7.* Training time and computational cost comparison across tasks. Time is measured in seconds (s), and memory in megabytes (MB). HSD total time includes one-time feature basis preprocessing time (indicated in parentheses).

| Model | Params | Magnetostatics | | Ext. Aerodyn. | | Toro. Transport | | Complexity |
| --- | --- | --- | --- | --- | --- | --- | --- | --- |
| | | Time | VRAM | Time | VRAM | Time | VRAM | |
| DeepONet | ∼240k | 6.7 | 369.5 | 7.5 | 369.8 | 4.5 | 171.1 | $\mathcal{O}(N)$ |
| FNO-3D | ∼230k | 177.0 | 382.7 | 29.5 | 382.7 | 36.2 | 194.4 | $\mathcal{O}(N \log N)$ |
| Geo-FNO | ∼250k | 49.6 | 383.9 | 49.6 | 384.0 | 35.6 | 173.5 | $\mathcal{O}(N \log N)$ |
| HSD (Ours) | ∼220k | 215.5* | 364.5 | 34.6* | 383.3 | 372.2* | 173.0 | $\mathcal{O}(Nk + N \log N)$ |
| GNO | ∼230k | 477.4 | 385.3 | 426.5 | 385.4 | 195.1 | 183.1 | $\mathcal{O}(N|E|)$ |
| MGN | ∼240k | 3983.0 | 382.7 | 1865.0 | 383.0 | 1129.3 | 174.4 | $\mathcal{O}(N|E|)$ |

*HSD time includes one-time offline spectral precomputation overhead (Magnetostatics: +57.1s, Ext. Aerodyn.: +1.6s, Toro. Transport: +3.8s).

We observe that DeepONet has the fastest training speed, but as discussed earlier, it exhibits lower physical and topological fidelity. MGN is extremely slow to train due to its explicit message passing mechanism, especially for larger mesh sizes. HSD demonstrates high efficiency in static field tasks, with slightly increased time in dynamic tasks due to the inclusion of the temporal residual correction module, but remains significantly faster than MGN and GNO.

## G.4. Computational Resources

All experiments were implemented using the PyTorch 2.9.0 framework, specifically utilizing the NVIDIA NGC PyTorch container (Release 25.09) for optimized performance. The computations were executed on high-performance computing nodes equipped with an AMD Ryzen Threadripper 7970X 32-core CPU and dual NVIDIA RTX PRO 6000 Blackwell Workstation Edition GPUs. Each GPU provides 96 GB of VRAM, totaling 192 GB of video memory, which provides sufficient capacity for high-resolution spectral field modeling. The software environment was built upon CUDA 13.0 to ensure optimal hardware acceleration. To guarantee experimental reproducibility, we employed a Docker-based containerization strategy to maintain consistent versions of all dependency libraries and drivers.

## G.5. Optimization and Training Configuration

All tasks employ the AdamW optimizer with a cosine annealing learning rate scheduler (CosineAnnealingLR) that gradually decays the learning rate from the initial value to zero. We designed corresponding loss functions for different physical field types: for vector field tasks (Magnetostatics and External Aerodynamics), the loss function is a weighted sum of flux loss $\mathcal{L}_{\text{flux}}$ and divergence loss $\mathcal{L}_{\text{div}}$, with weights set to $\lambda_{\text{flux}} = 1.0$ and $\lambda_{\text{div}} = 0.1$ respectively; for the scalar field task (Toroidal Transport), we use the standard $L^2$ relative error with an additional $L^1$ regularization term on sparse spectral coefficients. The weight decay coefficient is set to $10^{-4}$ for vector field tasks and $10^{-5}$ for the scalar field task.

## G.6. Model Hyperparameters

Table 8 summarizes the specific hyperparameter configurations for the HSD model and all baseline models across the three experimental tasks. To ensure fair comparison, the parameter counts for all models are controlled within a similar range (approximately 250k–300k).

Regarding hyperparameter selection, the following notes apply. In the External Aerodynamics task, the spectral truncation dimension $k$ is increased to 128 (compared to 64 for other tasks) to accommodate the significantly higher topological complexity and surface curvature variations of DrivAerNet++ geometries. In the Toroidal Transport task, the Fiber branch depth of HSD is increased to 6 layers, as the temporal evolution characteristics of high-frequency features in the advection-diffusion process require a deeper ambient space network to capture, whereas the static fields in Magnetostatics and External Aerodynamics tasks only require shallower network structures (4 and 3 layers, respectively).

*Table 8.* Hyperparameter configuration for each task.

| Hyperparameter / Model | Magnetostatics | Ext. Aero. | Toroidal Trans. |
|---|---|---|---|
| *Global Settings* | | | |
| Spectral truncation dimension $k$ | 64 | 128 | 64 |
| Batch size | 64 | 64 | 64 |
| Training epochs | 100 | 100 | 50 |
| Initial learning rate | $10^{-3}$ | $10^{-3}$ | $10^{-3}$ |
| *HSD (Ours)* | | | |
| Base branch layers | 2 (Gated MLP) | 2 (Gated MLP) | 2 (Gated MLP) |
| Base hidden dimension | 32 | 32 | 32 |
| Fiber branch modes | $4^3$ | $4^3$ | $4^3$ |
| Fiber hidden channels | 12 | 11 | 11 |
| Fiber depth | 4 | 3 | 6 |
| Fiber branch grid resolution | $16^3$ | $16^3$ | $16^3$ |
| *GNO* | | | |
| Hidden channels | 84 | 84 | 120 |
| Projection channels | 96 | 96 | 68 |
| Depth | 5 | 5 | 3 |
| Neighborhood radius | 0.2 | 0.2 | 0.15 |
| *FNO-3D* | | | |
| Modes | $4^3$ | $4^3$ | $4^3$ |
| Hidden channels | 21 | 21 | 20 |
| Depth | 2 | 2 | 3 |
| Grid resolution | $16^3$ | $16^3$ | $16^3$ |
| *MGN* | | | |
| Hidden dimension | 58 | 58 | 72 |
| Message passing layers | 10 | 10 | 8 |
| *DeepONet* | | | |
| Branch network layers | $[64, 64, 64]$ | $[64, 64, 64]$ | $[96, 96, 64]$ |
| Trunk network layers | $[64, 64, 64]$ | $[64, 64, 64]$ | $[68, 68, 64]$ |
| Basis function dimension $p$ | 74 | 74 | 64 |
| *Geo-FNO* | | | |
| Modes | 6 | 6 | 6 |
| Width | 12 | 12 | 9 |
| Depth | 2 | 2 | 4 |

## G.7. Training Curves

Figures 9, 10, and 11 show the training loss curves for each model across the three tasks. It can be observed that HSD exhibits stable decreasing trends in all three tasks and achieves the lowest final loss values. In contrast, FNO-3D and DeepONet show pronounced oscillations or premature plateauing in complex geometry tasks, while GNO and MGN, despite relatively stable training processes, still have higher final loss values than HSD. Notably, HSD maintains comparable training efficiency to other tasks in the External Aerodynamics task despite using a larger spectral truncation dimension.

# H. Additional Visualization Results

This section provides additional visualization samples for each task to more comprehensively demonstrate the prediction performance of HSD and baseline models across different test cases. All error maps use the same color scale mapping, with black to red to yellow to white indicating increasing error magnitude.

## H.1. Magnetostatics

Figures 12 and 13 show slice visualizations of magnetic vector fields for two additional test samples in the Magnetostatics task. It can be observed that under different geometric configurations and boundary conditions, HSD accurately captures the spatial distribution characteristics of the magnetic field, particularly maintaining low prediction errors in regions with large

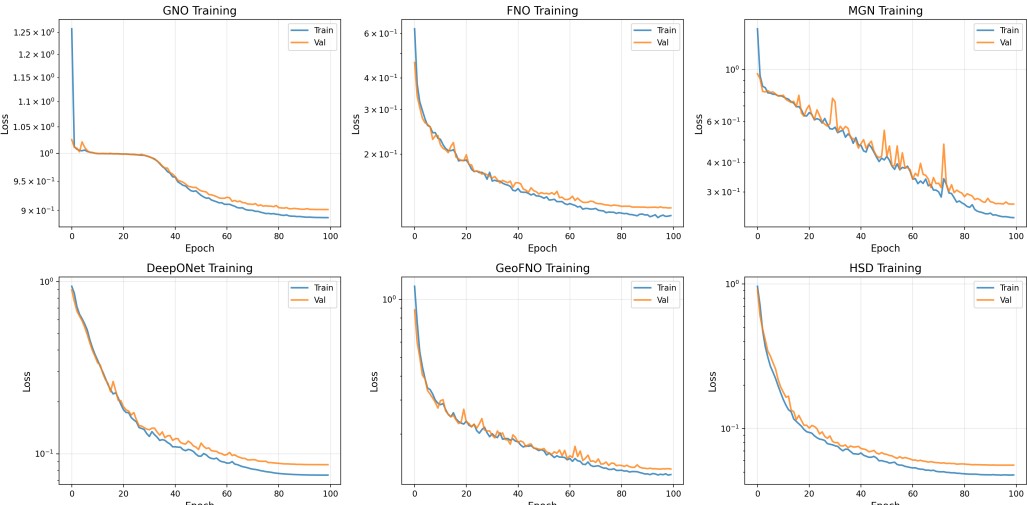

*Figure 9.* Training loss curves for all models on the Magnetostatics task. The horizontal axis represents training epochs, and the vertical axis shows MSE loss on a logarithmic scale.

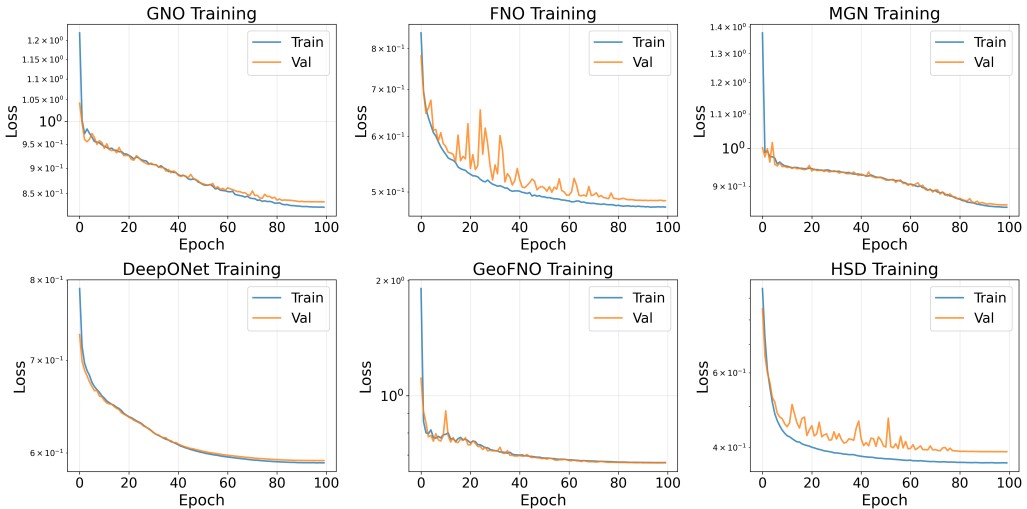

*Figure 10.* Training loss curves for all models on the External Aerodynamics task. The horizontal axis represents training epochs, and the vertical axis shows MSE loss on a logarithmic scale.

field strength gradients. In contrast, FNO-3D produces noticeable artifacts near complex boundaries due to its reliance on regular grid interpolation; while GNO and MGN can adapt to unstructured meshes, their field reconstruction accuracy in high-curvature regions remains inferior to HSD.

## H.2. External Aerodynamics

Figures 14 and 15 show velocity vector field prediction results for two different vehicle geometries in the External Aerodynamics task. These two samples represent streamlined and bluff body designs, respectively, corresponding to distinctly different flow field characteristics. HSD demonstrates accurate prediction capability for near-wall velocity distributions in both cases, particularly in regions with geometric discontinuities such as wheel arches and side mirrors, where errors are significantly lower than those of other methods. DeepONet, due to its point-wise evaluation limitations, struggles to capture spatially correlated flow field structures; Geo-FNO, despite introducing geometric transformations, still exhibits larger errors when handling vehicle geometries with high topological complexity in the DrivAerNet++ dataset.

Figures 16 and 17 further show surface flux distributions obtained by integrating the velocity field. As a key physical quantity

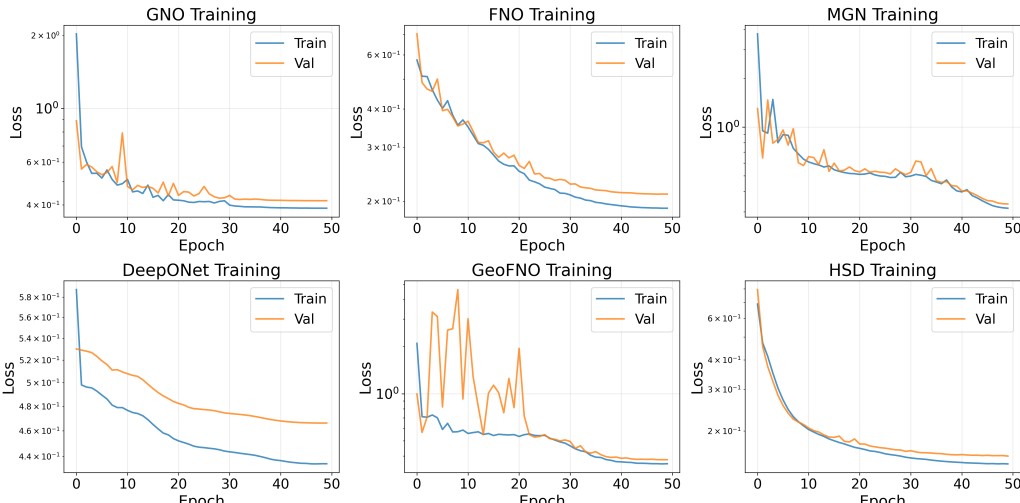

*Figure 11.* Training loss curves for all models on the Toroidal Transport task. The horizontal axis represents training epochs, and the vertical axis shows MSE loss on a logarithmic scale.

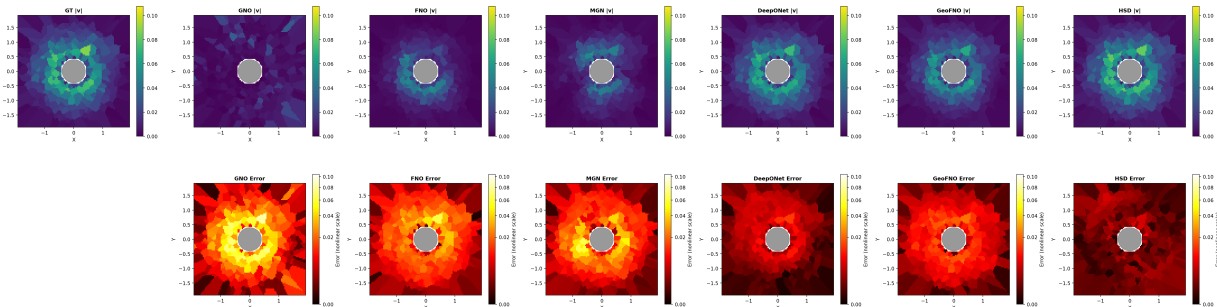

*Figure 12.* Slice visualization of magnetic vector field for the Magnetostatics task (Sample 1). Each column corresponds to a different model, with the top row showing predictions and the bottom row showing corresponding errors (color scale: black→red→yellow→white indicates increasing error).

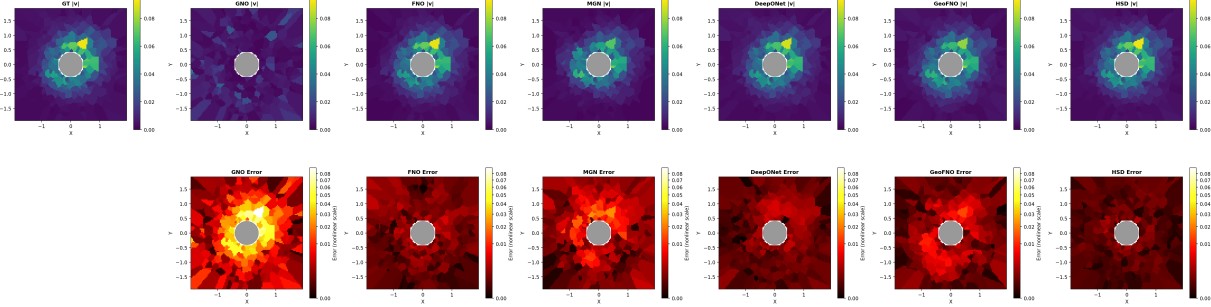

*Figure 13.* Slice visualization of magnetic vector field for the Magnetostatics task (Sample 2). Each column corresponds to a different model, with the top row showing predictions and the bottom row showing corresponding errors. Color scale same as Figure 12.

for downstream engineering analysis, flux requires higher prediction accuracy. It can be seen that the flux distribution predicted by HSD closely matches the Ground Truth, while baseline models exhibit pronounced spatial error accumulation effects in their flux predictions, especially in flow separation regions and wake confluence zones where errors are significantly amplified.

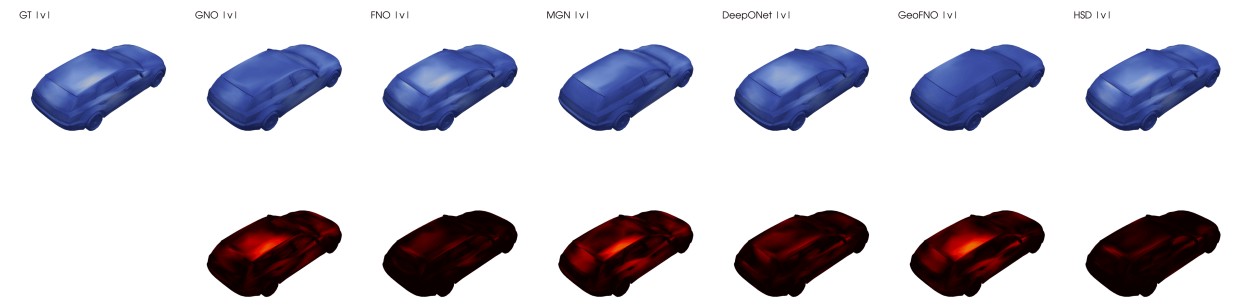

*Figure 14.* Velocity vector field prediction visualization for the External Aerodynamics task (Sample 1). Each column corresponds to a different model, with the top row showing predictions and the bottom row showing corresponding errors (color scale: black→red→yellow→white indicates increasing error).

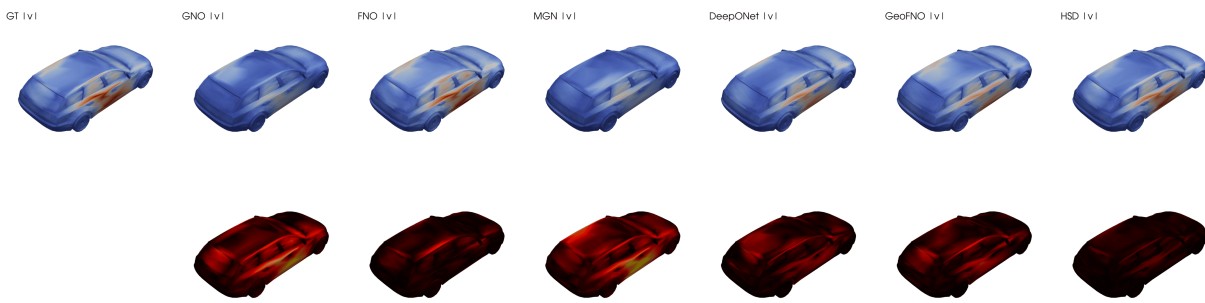

*Figure 15.* Velocity vector field prediction visualization for the External Aerodynamics task (Sample 2). Each column corresponds to a different model, with the top row showing predictions and the bottom row showing corresponding errors. Color scale same as Figure 14.

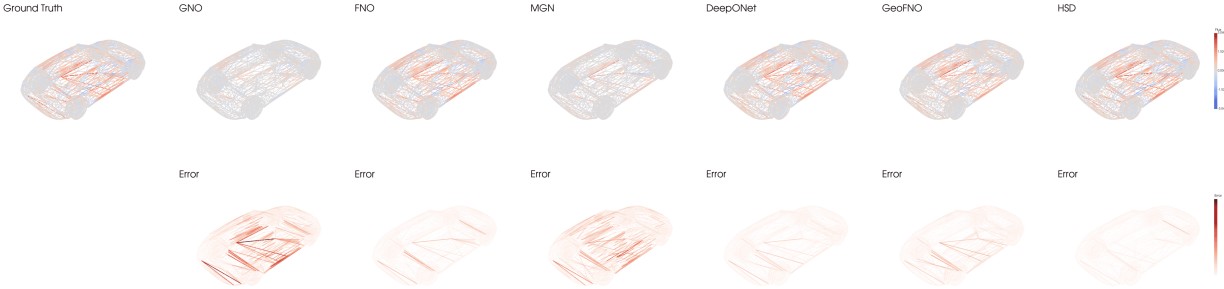

*Figure 16.* Surface flux prediction visualization for the External Aerodynamics task (Sample 1). Each column corresponds to a different model, with the top row showing predicted flux and the bottom row showing corresponding errors against Ground Truth. Color scale same as Figure 14.

### H.3. Toroidal Transport

Figures 18, 19, and 20 show three test samples with different initial conditions in the Toroidal Transport task. This task involves the advection-diffusion evolution of a scalar field on a torus, where the spatial frequency and localization of the initial field distribution directly affect the complexity of the final state. Under the low-frequency initial conditions shown in Figure 18, most models provide reasonable predictions; however, as the spatial structure complexity of the initial field increases (as in Figures 19 and 20), baseline model errors rapidly grow, manifesting as over-smoothing of high-frequency details or spurious oscillations. HSD, leveraging its spectral domain processing capability and geometry-aware architecture on fiber bundles, maintains consistently low error levels across all three samples, demonstrating its robust modeling capability for time-varying physical field evolution processes.

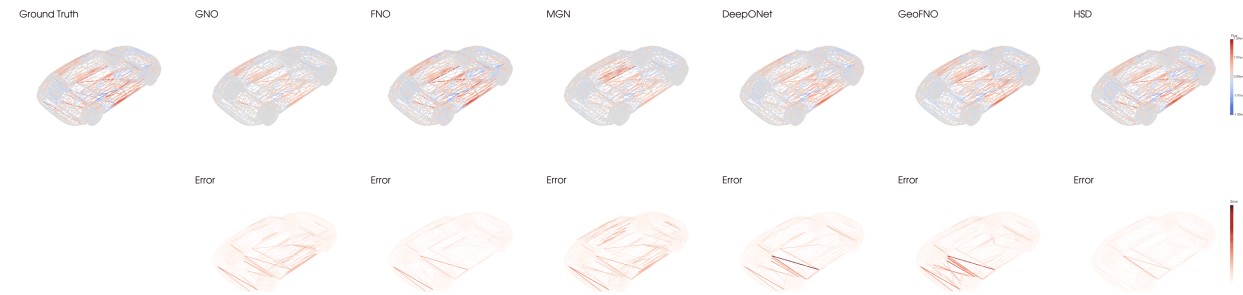

*Figure 17.* Surface flux prediction visualization for the External Aerodynamics task (Sample 2). Each column corresponds to a different model, with the top row showing predicted flux and the bottom row showing corresponding errors. Color scale same as Figure 14.

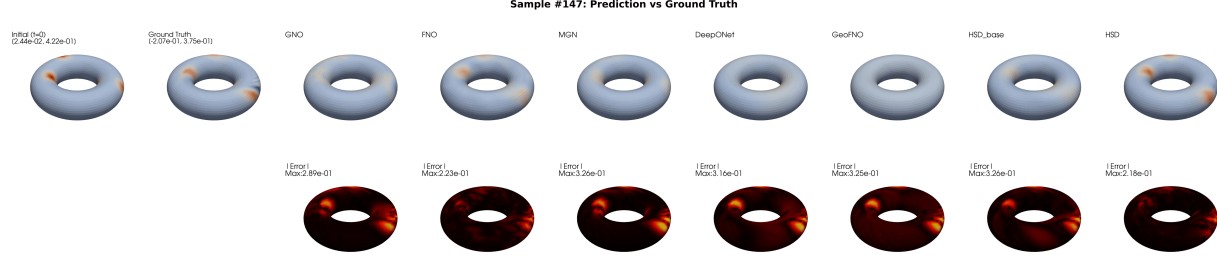

*Figure 18.* Scalar field prediction visualization for the Toroidal Transport task (Sample 1). From left to right: initial condition, Ground Truth, and prediction results from each model. The bottom row of each column shows corresponding errors (color scale: black→red→yellow→white indicates increasing error).

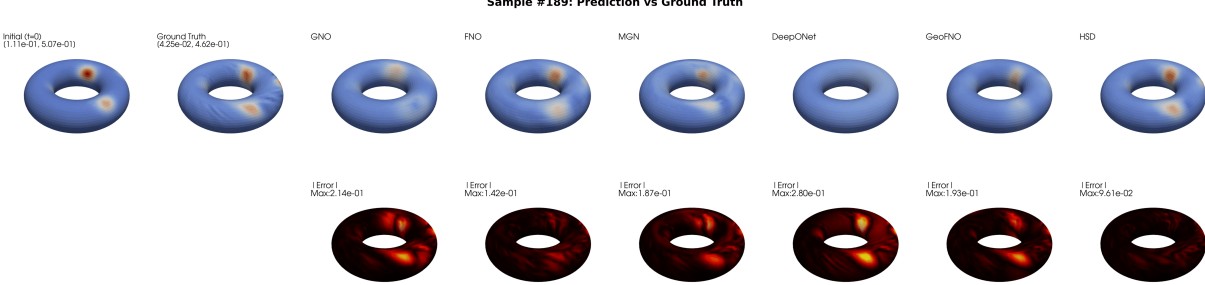

*Figure 19.* Scalar field prediction visualization for the Toroidal Transport task (Sample 2). From left to right: initial condition, Ground Truth, and prediction results from each model. The bottom row of each column shows corresponding errors. Color scale same as Figure 18.

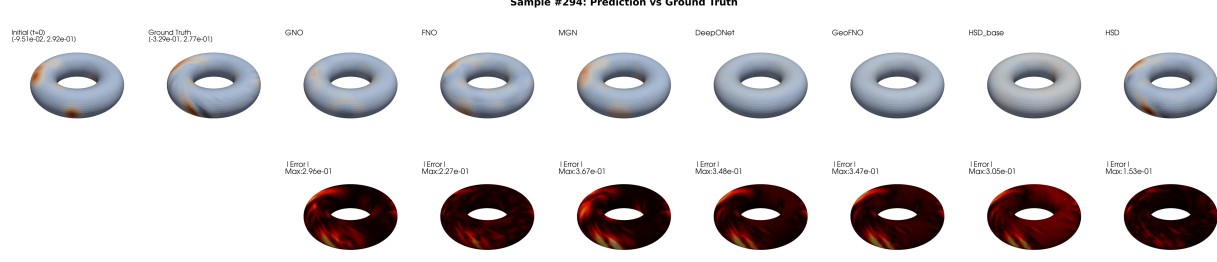

*Figure 20.* Scalar field prediction visualization for the Toroidal Transport task (Sample 3). From left to right: initial condition, Ground Truth, and prediction results from each model. The bottom row of each column shows corresponding errors. Color scale same as Figure 18.

