# OpenReview forum: "Topology-Preserving Neural Operator Learning via Hodge Decomposition"
_ICML.cc/2026/Conference — ICML 2026 regular_

### Official Review · Reviewer_4mwr · 2026-03-10

**Soundness:** 4
**Presentation:** 3
**Significance:** 4
**Originality:** 4
**Overall Recommendation:** 6
**Confidence:** 4

**Summary:**

The paper introduces the Hodge Spectral Duality approach to operator learning for PDEs on Riemannian manifolds. An orthogonal spectral decomposition of the Hodge Laplacian with respect to a corresponding inner product defines a fiber bundle supposed to represent low-dimensional global topological properties (base space) and high-dimensional local metric properties (fibers), respectively, of solutions. Two components of a corresponding decomposition of the solution operators are learned using numerical ground-thruth solutions and fused by another learned mapping for taking into account the non-commutativity of operators. Superior results in comparison to a range of other operator learning approaches are demonstrated in three experimental scenarios, including a base manifold with non-trivial topology.

**Compliance With Llm Reviewing Policy:**

Affirmed.

**Key Questions For Authors:**

Please adress my concern regarding the presentation in Appendices E and F.

Will code be provided?

**Limitations:**

yes

**Strengths And Weaknesses:**

Strenghts
This is a comprehensive and convincing attempt to achieve a “unifying” approach to operator learning, which copes with PDEs on arbitrary Riemannian manifolds and higher-order simplicial complexes for capturing cohomological structures and covering discretization in higher dimensions. The claim that the approach learns indeed the operator rather independently from the discrete representation is supported by numerical experiments. Details of the machine learning set-up (parameters, layers, learning curves etc.) are reported.

Weaknesses
I do not see a really weak point.

Regarding the presentation, as a reviewer not working in this area of research, I could follow very well the global structure (aspects of algebraic topology and Riemannian geometry) and the line of reasoning which explains why the approach should be superior to related work. Regarding further details, I get lost however, because they can only be understood by researchers in the field. For example, in Section 4 the operators mathcal{A}^k are indicated but not explicitly specified, nor is the decomposition (28) (at least in the appendix, this should be done). In the Apppendices E and F, I got lost regarding the indicated “high-level engineering”, e.g. switching to the ambient space and going back using both unknown concepts like Whitney reconstruction, known ones like kernel density estimation, and more. But how these methods and steps are combined is hard to decipher for readers outside the field.

Title: I would consider a formulation like "Neural Operator Learning on Manifolds using Hodge Spectral Duality" more to the point.

---

> ### Author Rebuttal · Authors · 2026-03-30
>
> Thank you very much for your suggestions.
>
>
> **Title Suggestion:**
>
> We agree that "Neural Operator Learning on Manifolds via Hodge Spectral Duality" more directly signals the core contribution and broadens accessibility. We are open to adopting this or a similar formulation in later revisions.
>
>
> **W1 / Q1 — Presentation of Appendices E and F:**
>
>
> We agree that the key objects deserve clearer exposition. We address each component directly.
>
> **Operator splitting.** $\mathcal{A}^k=\mathcal{A}^k_{\mathrm{Topo}}+\mathcal{A}^k_{\mathrm{Geom}}$ decomposes PDE operators into two complementary parts. $\mathcal{A}^k_{\mathrm{Topo}}$ is generated by $d,\delta,\Delta_k$ (the exterior derivative, codifferential, and Hodge Laplacian) and encodes homological and conservation constraints (e.g., $\nabla \cdot B = 0$, flux conservation through closed surfaces). $\mathcal{A}^k_{\mathrm{Geom}}$ is determined by the Riemannian metric $g$ and material parameters $\kappa$, and governs local diffusion, transport, and high-frequency geometric effects (e.g., anisotropic conductivity, spatially varying wave speed).
>
> **Fiber branch pipeline.** The fiber branch is the explicit sequence: $C^k(K)\xrightarrow{\iota}L^2(\Omega_{\mathrm{aux}})\xrightarrow{\mathrm{FFT}}L^2(\Omega_{\mathrm{aux}})\xrightarrow{R}C^k(K)\xrightarrow{I-\Pi^k_{\mathrm{base}}}V^k_{\mathrm{fiber}}$. Here $\iota$ performs Whitney interpolation followed by KDE smoothing. Whitney maps lift discrete cochains (edge/face values) to continuous piecewise $k$-forms that respect the simplicial complex's structure and orientation, for 1-forms, the line integral of the lifted field along each oriented edge recovers the original edge coefficient. KDE smooths derivative jumps at element boundaries so that the lifted field satisfies band-limited conditions required for stable FFT on the auxiliary Euclidean grid. $R$ reverses this: grid interpolation followed by Whitney projection pulls back the ambient field to a discrete cochain. The final projection $(I-\Pi^k_{\mathrm{base}})$ restricts output to the fiber subspace, ensuring geometric updates cannot pollute topological components.
>
> A spatial encoding ablation validates this design across all tasks including two challenging additions: **Ellipsoid Aero** (nonlinear flow, genus-0) and **Torus Helmholtz** (Helmholtz equation, genus-1).
>
> | Spatial Encoding | Ellipsoid Aero (Relative L2) | Torus Helmholtz (Relative L2) |
> |---|---|---|
> | Whitney-KDE (Ours) | **0.037** | **0.058** |
> | Raw Euclidean 3D | 0.042 | 0.065 |
>
> **Commutator corrector (Appendix F).** $C_\theta^{(\ell)}$ addresses the fundamental issue that $\mathcal{A}^k_{\mathrm{Topo}}$ and $\mathcal{A}^k_{\mathrm{Geom}}$ do not generally commute. By the Leibniz rule, the residual of $[\mathcal{A}^k_{\mathrm{Topo}}, \mathcal{A}^k_{\mathrm{Geom}}]$ is driven by coupling terms between the material parameter gradient $\nabla\kappa$ and the field gradient $\nabla u$, e.g., $[\Delta, M_\kappa]u = (\Delta\kappa)u + 2\langle\nabla\kappa, \nabla u\rangle_g$. $C_\theta^{(\ell)}$ takes the cross-features $z^{(\ell)}$ (which simultaneously encode spectral derivative proxies of the field $M_d^{(k)}c, M_\delta^{(k)}c$ and geometric embedding information) as input. Outputs is restricted to the fiber subspace via $(I-\Pi^k_{\mathrm{base}})$, ensuring that the correction does not pollute the topological components.
>
> **Q2 — Code Availability:**
>
> The anonymous repository is in Appendix A, containing training scripts, configurations, and data preprocessing for all experimental tasks. After double-blind review, we will release a PyPI library providing a unified API supporting mesh, point cloud, and graph inputs, enabling researchers to apply HSD to new problems without configuring algebraic topology infrastructure from scratch.

---

> > ### Author Rebuttal · Reviewer_4mwr · 2026-04-01
> >
> > Thanks for your response. This is excellent work. My corresponding rating remains unchanged.

---

> > > ### Author Response · Authors · 2026-04-06
> > >
> > > We sincerely thank you for your positive feedback and support, and greatly appreciate your recognition of the quality and contribution of our work.

---

### Official Review · Reviewer_R3sF · 2026-03-11

**Soundness:** 3
**Presentation:** 3
**Significance:** 3
**Originality:** 4
**Overall Recommendation:** 5
**Confidence:** 4

**Summary:**

To solve PDEs on manifolds with neural operator, the paper introduces a novel dual branch architecture to treat the low-frequency, global, topological aspect and the high-frequency, local, geometry features of the manifold separately, with Hodge operator-induced inner product to define orthogonal projections to keep the two streams separate. To take into account the nonlinearity, they introduce a commutator correction third branch within the high-frequency channel, hence preserving topological features from layer to layer. Experiments show improvements from a number of baseline models.

**Compliance With Llm Reviewing Policy:**

Affirmed.

**Final Justification:**

I will maintain my score, this work is clearly valuable. Though I still suggest the authors can improve the presentation to better present the formalism and motivating mathematical theories like the topology-geometry dichotomy, providing better intuition for the broader audience.

**Key Questions For Authors:**

1. Why is the gMLP_k chosen in this specific way? Why the only nonlinearity is applied to the gating stream? Why no bias inside the gating? For example, if c_k^{(l)} = 0 from the layer below,  the whole layer of equation (5) seems to be degenerate. Is this intended?
2. Why use Whitney + KDE in the fiber branch? Does this introduce any undesirable inductive biases?
3. From Table 1, the best-performing baselines are FNO-3D and DeepONet, neither is sensitive to the topology of the manifold in the standard implementations. Is this what you expect? Does this indicate insufficient hyperparameter tuning of baseline models or that the benchmarks are insufficiently complex in topology?
4. Have you done ablation studies of the gMLP_k with standard multilayer MLPs?
5. What are the input/output of each of the experiment tasks more precisely? Do they fit in the architecture (and formalism as in equation (1)) in a unified way? Is there any limiation of this architecture for different types of PDEs?

**Limitations:**

Yes.

**Strengths And Weaknesses:**

Strengths:
- The dual branch method is well motivated and theoretically principled, genuinely novel application of well-understood mathematical phenomenon.
- The architecture is carefully designed to optimize both mathematical interest (topological preservation of harmonic modes) and expressivity.
- The experiments show overwhelming improvements from baselines over three well-chosen tasks continguous to related literature (neural operator on topologically nontrivial manifolds).
-  Decent analysis of spectral bias and computational complexity

Weaknesses:
- The appendix is insufficient for audience not familiar with the background theory. Missing: PDE in differential form formulation; the connection between equation (2) and equation (1), the topology-geometry dichotomy in relation to equation (2), (1), and (3), to justify the dual branch approach.
- The formalism is inaccurate and not sufficiently general. Is (1) meant to be time-dependent or with generic boundary conditions? Will other data than $f_k$ figure in the input of the neural operator?
- The architecture presentation is not clear: how are the initial layers assigned with any values? For example, when you preserve the harmonic modes from a lower level, how are they initialized - from $f_k$ (which is unlikely)?
- Insufficient theoretical derivation and justification about the desiderata: topology and expressivity.
- The implementation is fragmented without a unified architecture, has to be handcrafted from problem to problem.

---

> ### Author Rebuttal · Authors · 2026-03-30
>
> Thank you for your emphasis on theoretical foundations.
>
> **W1: Insufficient Appendix:**
>
> Differential-form PDEs take the form $\frac{\partial u_k}{\partial t} + A_k(g,\kappa;d,\delta)u_k = f_k$, where $A_k$ involves $d,\delta,\Delta_k{=}d\delta{+}\delta d$ and metric/material terms. Eq. (1) states the operator learning objective; Eq. (2) decomposes $A_k = A_k^{\mathrm{Topo}} + A_k^{\mathrm{Geom}}$: $A_k^{\mathrm{Topo}}$ from $d,\delta,\Delta_k$ (metric-free homological constraints) and $A_k^{\mathrm{Geom}}$ from the metric $g$ and material $\kappa$ (diffusion, transport). Appendix C defines $d,\delta,\Delta_k$ in DEC; Appendix D defines base/fiber subspaces motivating Eq. (3).
>
> Example: Maxwell's equations ($\nabla\cdot B=0,\ \nabla\times E=-\partial_t B$, etc.) encode as $dF=0,\ \delta F=J$ where $F=B+E\wedge dt$: $d$ captures metric-free global topology; $\delta$ encodes metric/local geometry.
>
> **W2: Generality of Formalism:**
>
> Our framework is designed to capture differential-form PDEs. Eq. (1) covers steady-state $A_k(u_k;g,\kappa,\partial M,f_k)=0$ with $G_k:(f_k,u_k|_{\partial M},\kappa)\mapsto u_k$, and time-varying $\partial_t u_k + A_k=0$ with $G_k$ as an evolution operator. Inputs include source terms, boundary/initial conditions, and material coefficients. The five experimental tasks span steady-state/time-varying regimes and show the formalism's coverage across form degrees.
>
> **W3/W4: Initialization & Guarantees:**
>
> Harmonic modes at each layer come from spectral projection $c_k^{(\ell)}=\Phi_k^\top \star_k \omega_k^{(\ell)}$, with $\omega_k^{(0)}$ initialized from $f_k$. Preserving harmonic modes means replacing $\tilde c_k^{(\ell)}$ at harmonic indices with the original $c_k^{(\ell)}$ after each update, preventing destruction of cohomological content. The projection $\Pi^k_{\mathrm{base}}$ is computed from eigenvectors of $\Delta_k$; replacement is exact under the discrete Hodge inner product. The topology-preserving guarantee follows from $\ker \Delta_k \cong H^k(M;\mathbb{R})$ [1].
>
> For expressivity, the Fiber branch admits a universal approximation result: any continuous operator on $V^k_{\mathrm{fiber}}$ can be approximated, since the ambient FFT + neural operator satisfies universal approximation conditions [7]. HSD trades unrestricted expressivity for guaranteed topological invariance, which is a principled tradeoff for conservation-law PDEs.
>
> **W5/Q5: Implementation & Input-Output Spec:**
>
> HSD is ready for packaging as a PyPI library where all tasks share one backbone: $G^k: C^r(K,\mathbb{R}) \to C^k(K,\mathbb{R})$, supporting form-degree combinations $0{\to}0$, $0{\to}1$, $1{\to}0$, $0{\to}2$, etc. Task-specific inputs are the mesh, form degree $k$, and training data; $\Phi_k, B_k, *_k$ are derived automatically via DEC. The architecture suits conservation-law and curl/divergence-constrained PDEs. It maybe less suited for strongly nonlocal integral operators, or moving interfaces with time-varying topology.
>
> **Q1/Q4: gMLP Design & Comparison**
>
> gMLP is a pseudo-spectral bilinear layer: $\varphi(W_g q) \odot (W_c q)$ approximates quadratic nonlinearities (e.g., $u \cdot \nabla u$) via low-rank bilinear interaction. Zero gate bias enforces origin homogeneity ($c_k^{(\ell)}{=}0 \Rightarrow$ zero output), consistent with zero state producing no nonlinear energy. We validate on 2 additional tasks:
>
> | Layer | Ellipsoid Aero (rel. L2) | Torus Helmholtz (rel. L2) |
> |---|---|---|
> | MLP | 0.037 | 0.056 |
> | **gMLP** | **0.026** | **0.049** |
>
> **Q2: Whitney + KDE in Fiber Branch:**
>
> Whitney interpolation lifts discrete cochains to continuous fields consistent with simplicial structure; KDE smooths derivative jumps at cell boundaries for stable FFT on the auxiliary Euclidean grid, preserving discrete exterior derivative structure and local orientation. Fiber branch output is projected via $(I-\Pi^k_{base})$ to prevent geometric bias from polluting topological components.
>
> |  | Ellipsoid Aero | Torus Helmholtz |
> |---|---|---|
> | Whitney-KDE | **0.037** | **0.058** |
> | Raw Euclidean 3D | 0.042 | 0.065 |
>
> **Q3: Strongest Baselines:**
>
> Yes, this is expected. FNO's global frequency-domain convolution is advantageous with full-field observability [2]; local message-passing (MGN, GNO) bottlenecks under global coupling [3]. We added two topology-demanding tasks and three attention-based baselines:
>
> |  | Ellipsoid Aero (rel. L2) | Torus Helmholtz (rel. L2) |
> |---|---|---|
> | **HSD (Ours)** | **0.037** | **0.058** |
> | GNOT (2023) [4] | 0.144 | 0.277 |
> | ONO (2023) [5] | 0.155 | 0.262 |
> | HAMLET (2024) [6] | 0.159 | 0.250 |
> | DeepONet | 0.221 | 0.652 |
> | GeoFNO | 0.240 | 0.449 |
> | FNO | 0.246 | 0.418 |
> > [1] Hatcher, A. Algebraic Topology. Cambridge Univ. Press, 2002.
> > [2] Li et al., FNO, ICLR 2021.
> > [3] Sanchez-Gonzalez et al., MGN, ICMLW 2022.
> > [4] Hao et al., GNOT, ICML 2023.
> > [5] Raonić et al., ONO, NeurIPS 2023.
> > [6] Bryutkin et al., HAMLET, ICML 2024.
> > [7] Chen & Chen, IEEE TNN 1995

---

> > ### Author Rebuttal · Reviewer_R3sF · 2026-04-04
> >
> > Thanks for the new experiments which are very assuring. I have one question: the \omega_k is initialized from f_k but you preserve the harmonic mode from the initial layer, hence from f_k. But f_k is the forcing term, as opposed to the initial condition in the time-dependent setting (this is why I was asking about the generality of the formalism and the notation). Is it still to be preserved exactly through all layers? More theoretical explanation of this will be helpful.

---

> > > ### Author Response · Authors · 2026-04-05
> > >
> > > Thank you for raising this question. We agree that, for time-dependent PDEs, preserving the harmonic component of the solution is different from copying the harmonic component of the forcing term. We appreciate you pointing out that the notation in our previous rebuttal and in Eq. (1) caused confusion, as the phrasing was relatively loose and ambiguous.
> > >
> > > We would like to clarify that, in our current dynamic experiment, the learned map is from the initial condition $u_0$ to the trajectory $u(t)$. The initial state uses the generic notation $\omega_k^{(0)}$, which is constructed from the problem data ($f_k$, boundary conditions, material parameters, etc.) according to the requirements of each task. In the revision, we will specify the initialization details on a per-task basis: $\omega_k^{(0)}$ is respectively the discrete 1-form representation of the vorticity field, the initial magnetic field approximation constructed from the source field via a Poisson pre-solve, and the initial concentration distribution $u_0$ itself across the three tasks. Since these initializations are all constructed from the problem data that uniquely determine the harmonic class (boundary conditions and topological constraints), Hodge theory guarantees that their harmonic components are consistent with those of the true solution.
> > >
> > > The unified mechanism provided by HSD is as follows: topology determines the harmonic subspace $\ker(\Delta_k)$; the specific PDE state determines the coefficients in that subspace. Accordingly, the role of our harmonic projection is to prevent the fiber/geometric branch from contaminating the harmonic channel, preserving the harmonic component of the reference field, thereby decoupling the global topological invariants of the target solution from the metric-driven local dynamics.
> > >
> > > We agree that the formalism for the time-dependent case should be written more explicitly. In the revision, we will state the operator as $\mathcal{G}\_{\Delta t}(u\_t, f\_t, b\_t, \kappa) \mapsto u\_{t+\Delta t}$, and clarify the scope of the current harmonic hard-constraint formulation: it is appropriate when the relevant harmonic class is supplied by the initial/boundary data and remains fixed over the modeled interval. We will also clarify that the current form of HSD does not cover the most general forced time-dependent PDEs, in which harmonic coefficients may be injected or changed over time. For the steady-state setting, the harmonic projection mechanism should be understood as constraining the network's search to the admissible harmonic class determined by the problem data.

---

### Official Review · Reviewer_eS1d · 2026-03-11

**Soundness:** 3
**Presentation:** 4
**Significance:** 3
**Originality:** 3
**Overall Recommendation:** 5
**Confidence:** 3

**Summary:**

In "Topology-Preserving Neural Operator Learning via Hodge Decomposition", the authors propose a novel approach to learning neural operators working on discretised differential forms on simplicial complexes.
The key idea of the method is to learn two different operators for the low-frequency (corresponding to topological and global geometric features) and high-frequency (corresponding to local features) parts of the spectrum of the Hodge Laplacian.
While the key part of the low-frequency uses the Hodge Laplacian eigenbases, the high-frequency part is learned by lifting the input to the ambient Euclidean space, applying a neural operator there, and pulling back to the differential forms.
The authors benchmark there methods agains several baselines for neural operators on a number of different tasks.

**Compliance With Llm Reviewing Policy:**

Affirmed.

**Final Justification:**

The rebuttal clarified my remaining questions. Hence, I see no reason to deviate from my positive assessment.

**Key Questions For Authors:**

1. As mentioned above, the curl eigenvectors of the k-HL correspond to the gradient eigenvectors of the (k+1)-HL, and are related by the boundary and co-boundary operators. Thus, the method seems to duplicate these information. Could you please clarify whether there is a direct reason to do this, and provide an ablation study for using only the harmonic part of the spectrum, as well as the curl and gradient parts separately (obtained by applying the boundary and co-boundary operators to the low-frequency part of the spectrum), and only using the relevant curl and gradient eigenvectors in the spectral decomposition in dimensions k-1 and k+1?
2. The background section of the paper uses the Hodge star operator as a map between cochains on k-forms and cochains on k-forms. This does not agree with the definition from the literature, where the Hodge star maps k-forms to (n-k)-forms. Could you please clarify this point?
3. In the situation where there are multiple harmonic eigenvectors, how does the method distinguish between them and how do you pick the correct basis?
4. I understand that section 3 contains a lot of maths, however, I have been unable to locate the definition of the operator G_fibre, as it is implemented in the computations. Could you please clarify where this is defined?
5. What is the training and inference time of the method compared to the baselines on all of the experiments?

**Limitations:**

yes

**Strengths And Weaknesses:**

Strengths:
* The paper is well-written, the key intuition behind the method is clearly explained in the main text, and the rich appendix provides more necessary details on mathematics, implementation and experiments.
* The idea to use the Hodge spectrum to separate the low- and high-frequency components of the signal, and propose two different operators for them is a great idea and novel.
* In the experiments, the method is carefully evaluated on a number of different tasks and a number of different baselines, and the results are highly favourable.
* The theoretical time complexity of the method is shown to be better than the baselines after computing the Hodge Laplacian eigenbases.

Weaknesses:
* The most modern baseline the method is compared against is from 2021, which leaves the question of how the method performs against methods from the last four years.
* Although the title of the paper includes Hodge decomposition, the method only partially makes use of this: When G_base computes the low-frequency part c_k of the signal, it does not distinguish the harmonic, curl and gradient components of the signal, but rather stores the components in a a single vector ordered by eigenvalue. The curl eigenvalues of the Hodge Laplacian of dimension k and the gradient eigenvalues of the HL in dimension k+1 coincide, with the eigenvectors being related by applying the exterior derivative/boundary operator. Thus, the method essentially duplicates the same information by applying the boundary and co-boundary operators, instead of using the Hodge decomposition to directly separate the signal. (Please correct me if I have misunderstood this point.)
* The computation of the Hodge Laplacian eigenbases limits the method to small simplicial complexes.

---

> ### Author Rebuttal · Authors · 2026-03-30
>
> Thank you for your suggestions.
>
> **W1: Modern Baselines:**
>
> Our baselines were selected to cover classical design principles: graph message passing, spectral convolution, and branch-trunk regression. We now extend to GNOT [1], ONO [2], and HAMLET [3], with matched parameter counts.
>
> Our method outperforms each of these baselines on our existing tasks and two new and more challenging tasks: **Ellipsoid Aero** (nonlinear flow, genus-0) and **Torus Helmholtz** (Helmholtz equation, genus-1).
>
> |  | Ellipsoid Aero (rel. L2) | Torus Helmholtz (rel. L2) |
> |---|---|---|
> | **HSD (Ours)** | **0.037** | **0.058** |
> | GNOT (2023) [1] | 0.144 | 0.277 |
> | ONO (2023) [2] | 0.155 | 0.262 |
> | HAMLET (2024) [3] | 0.159 | 0.250 |
> | DeepONet | 0.221 | 0.652 |
> | GeoFNO | 0.240 | 0.449 |
> | FNO | 0.246 | 0.418 |
>
> **W2/Q1: Partial Use & Encode Redundancy:**
>
> We would like to clarify that $\mathcal{M}$\_d$^{(k)}c_k$ and $\mathcal{M}$\_δ$^{(k)}c_k$ do not re-encode the same information. The images of $d_k$ and $\delta_k$ lie in different spaces corresponding to $(k+1)$-order features (curl/circulation) and $(k-1)$-order features (divergence/potential structure) respectively. These are distinct subspaces of the de Rham complex.
>
> We clarify that the implementation does not use three fully independent parameterized branches. The static Hodge decomposition holds pointwise, but exact and coexact components are cross-coupled under nonlinear operators. We concatenate $q_k^{(\ell)}=\mathrm{concat}(c_k^{(\ell)}, M_d^{(k)}c_k^{(\ell)}, M_\delta^{(k)}c_k^{(\ell)})$ and pass through a pseudo-spectral bilinear layer gMLP to learn nonlinear mode mixing.
>
> We conducted the ablation of isolating exact, coexact, and harmonic components via boundary/co-boundary operators:
>
> | | Full Hodge | Exact only | Coexact only | Harmonic only |
> |---|---|---|---|---|
> | Ellipsoid | **0.084** | 0.113 | 0.338 | 1.002 |
> | Torus | **0.475** | 0.995 | 0.483 | 1.003 |
>
> **W3: Scalability:**
>
> HSD scalability has three components. (1) *Offline preprocessing*: DEC assembly is $O(N_k)$; shift-invert Lanczos extracts $m_k$ eigenpairs in $O(N_k \cdot m_k)$ on the sparse Hodge Laplacian (Appendix G). (2) *Online inference*: the forward pass operates in the $m_k$-dimensional spectral space with compression ratio $m_k/N_k$ (e.g., 128/$10^5$), giving $O(m_k)$ per-layer cost via GPU-friendly GEMM. (3) *Coarse-to-fine transfer*: Sec. 4.8 shows low-frequency topological modes are stable under remeshing, enabling transfer without recomputing the spectral basis. In wall-clock time, HSD is competitive with or faster than baselines even including eigendecomposition preprocessing (Table 6).
>
> **Q2: Definition of the Hodge Star:**
>
> The continuous Hodge star operator is defined $\star:\Omega^k(M)\to\Omega^{n-k}(M)$. In our discrete implementation (DEC [5]), $\star_k: C^k(K,\mathbb{R})\to C^k(K,\mathbb{R})$ is the mass matrix inducing the discrete Hodge inner product on $C^k(K,\mathbb{R})$: $\langle \alpha,\beta\rangle_{\star_k}=\alpha^\top \star_k \beta$, and $\delta_k=\star_{k-1}^{-1}B_k \star_k$ (Appendix C.2). This is the matrix representation of the $k\to n{-}k$ Hodge star after discretization [4,6].
>
> **Q3: Basis Choice:**
>
> When the harmonic eigenspace has dimension greater than 1, no canonical basis exists; the geometrically meaningful object is the entire subspace $ker L_k$. We use the generalized eigendecomposition of the discrete Hodge Laplacian to obtain Hodge-orthonormal harmonic modes (Appendix D); different bases are merely coordinate changes within this subspace and do not alter cohomology classes or the topological content represented there.
>
> **Q4: Definition of $G_{fiber}$:**
>
> $G_{fiber}: C^k(K) \to V^k_{\mathrm{fiber}}$ maps a $k$-cochain to its geometric update via $C^k(K) \xrightarrow{\iota} L^2(\Omega_{\mathrm{aux}}) \xrightarrow{\mathrm{FFT/NN}} L^2(\Omega_{\mathrm{aux}}) \xrightarrow{R} C^k(K) \xrightarrow{I-\Pi^k_{\mathrm{base}}} V^k_{\mathrm{fiber}}$, where $\iota$ is a Whitney+KDE lifting, the middle step is a frequency-domain neural operator, and $R$ is grid interpolation + Whitney projection. The final $(I-\Pi^k_{\mathrm{base}})$ restricts the output to $V^k_{\mathrm{fiber}} = (V^k_{\mathrm{base}})^{\perp_{*_k}}$. Eq. (9) combines this with the commutator correction. We will clarify this in our revision.
>
> **Q5: Training and Inference Time:**
>
> Training time comparisons are positively correlated with inference time and reported in Sec. 4.7 and Appendix H.2/Table 6. The revision includes explicit inference-time metrics across all five tasks and timing data for larger-scale experiments.
>
> > [1] Hao et al., GNOT, ICML 2023.
> > [2] Raonić et al., CNO, NeurIPS 2023.
> > [3] Bryutkin et al., HAMLET, ICML 2024.
> > [4] Hatcher, Algebraic Topology, Cambridge Univ. Press, 2002.
> > [5] Hirani, Discrete Exterior Calculus, PhD Thesis, Caltech, 2003.
> > [6] Lim, Hodge Laplacians on Graphs, SIAM Review, 2020.

---

> > ### Author Rebuttal · Reviewer_eS1d · 2026-04-01
> >
> > Thanks for the rebuttal, I will keep my positive score.

---

> > > ### Author Response · Authors · 2026-04-06
> > >
> > > Thank you very much for your follow-up and for maintaining your positive score. We appreciate your acknowledgment that our rebuttal has resolved your concerns.

---

### Official Review · Reviewer_BsYS · 2026-03-12

**Soundness:** 4
**Presentation:** 2
**Significance:** 3
**Originality:** 3
**Overall Recommendation:** 4
**Confidence:** 2

**Summary:**

This work proposed a structure-preserving learning framework for learning on manifolds. The proposed method is supposed to preserve the known topological structure for a given manifold.

**Compliance With Llm Reviewing Policy:**

Affirmed.

**Final Justification:**

This work is well-written and has rigorous math background.

**Key Questions For Authors:**

Does this method require the geometry of the object to be known in advance?

Can this method explore topological invariants from data?

Does this spectral decomposition guarantee all topological invariants contained in the base part?

**Limitations:**

I suggest that the authors to add one section for the Hodge Spectral Duality theorem. The connection between the topological invariants and the vector fields is not so obvious.

**Strengths And Weaknesses:**

The proposed method is mathematically sound.

The presentation needs to improve. It is not reasonable to assume that all readers are familiar with the Hodge Spectral Duality theorem.

This work addressed a specific need for structure preservation for learning fields on a manifold.

This work is novel according to my limited expertise.

---

> ### Author Rebuttal · Authors · 2026-03-30
>
> Thank you for the thoughtful review and constructive questions.
>
> **W1 — Accessibility and Presentation:**
>
> We agree clarity for a broad audience is essential. Appendix B gives a linear-algebra primer for the Hodge Decomposition. To make this concete, we will connect standard PDE forms to Hodge-de Rham Decomposition $\Omega^k = \mathrm{im}(d) \oplus \mathrm{im}(\delta) \oplus \ker(\Delta_k)$ in Sec. 3, and explain why the unique orthogonal decomposition of arbitrary $k$-forms on compact oriented Riemannian manifolds serves as an effective inductive bias for physical systems [1] (see W1 for Reviewer R3sF).
>
>
> In addition, we will release a PyPI library with a unified interface for mesh, point cloud, and graph inputs after double-blind review.
>
> **Q1 — Does the method require prior knowledge of geometry?**
>
>
> Yes, we assume the input is a mesh. This approach is standard when solving PDEs on manifolds [2].
>
> HSD is, however, robust to imperfect geometric information. The pseudo-spectral bilinear layer on the geometry-derived spectral basis learns spectral coefficient evolution, showing clear potential for data-driven sparse spectral mode selection and topological subspace identification.
>
>
> To demonstrate this, we now include additional experiments comparing Laplacian reconstruction for mesh, graph, and point cloud (via KNN and heat kernels [3,4]) inputs. We perform this experiment for two complex tasks: **Ellipsoid Aero**, nonlinear flow with vortex-source pairs and global moment coupling on a genus-0 surface; **Torus Helmholtz**, Helmholtz equation with two independent harmonic 1-forms on a genus-1 torus.
>
> | Input Format | Ellipsoid Aero (rel. L2) | Torus Helmholtz (rel. L2) |
> |---|---|---|
> | Mesh | 0.036 | 0.054 |
> | Point Cloud | 0.039 | 0.057 |
> | Graph | 0.036 | 0.056 |
>
>
> **Q2 — Can the method discover topological invariants from data?**
>
> Thank you for raising this exciting direction. Harmonic forms satisfy $\ker \Delta_k = \ker(d_k) \cap \ker(\delta_k)$, which is the precise algebraic characterization of topological non-triviality. The nullspace dimension of $\Delta_k$ equals the $k$-th Betti number, constituting the algebraic realization of all $k$-th order topological channels on the manifold [5]. HSD covers computation and optimization of this realization, enabling topological subspace identification.
>
> While we have yet to use this to discover topological invariants, we did explore a proof-of-concept Topological Data Analysis application: HSD achieves 91.0% accuracy on point-cloud Betti number prediction, outperforming the Euclidean distance baseline at 84.6%, with a 17.6% lead on higher-order features at $b_1=2$.
>
> **Q3 — Are topological invariants in the Base branch?**
>
> Yes, topological invariants are encoded by the harmonic k-forms in the Base branch. By the Hodge theorem, there is a dimension-exact isomorphism $\ker \Delta_k \cong H^k(M;\mathbb{R})$, which is the exact algebraic realization of the manifold topology [6]. The harmonic part of the Base branch operates within this topologically equivalent subspace.
>
> The Fiber branch which encodes the geometry is confined to the $L^2$-orthogonal complement of the Base branch. Since the Hodge inner product between these subspaces is zero, the features learned by the Fiber branch cannot alter the harmonic coefficients in the Base branch.
>
> **L1 — Connection between topological invariants and vector fields:**
>
> To clarify this connection, we will add a new subsection introducing Hodge-de Rham Decomposition and its relationship to physical systems.
>
> Specifically, in physical systems, macroscopic conserved quantities of vector fields, net circulation around holes, net flux trapped in cavities, are mathematically equivalent to harmonic forms spanning $\ker \Delta_k$ [7].
>
> For example, Maxwell's equations split into two classes: $dF = 0$ is a purely topological constraint stating $F$ is closed, so magnetic flux through any closed surface is conserved independently of the metric or source $J$; $d{\star}F = J$ is a geometric constraint governing local dynamics via the Hodge star ${\star}$.
>
> Magnetic flux conservation $\oint B \cdot dA = \text{const}$ corresponds to the cohomology class $[F] \in H^2(M)$ characterized by $\beta_2$. Standard neural operators minimizing only pointwise error risk violating physical conservation laws; our Base branch explicitly preserves topological invariants.
>
> > [1] Bhatia et al. The Helmholtz-Hodge Decomposition—A Survey, 2013.
> > [2] Quarteroni, A. Numerical Models for Differential Problems, 2009.
> > [3] Belkin et al. Constructing Laplace Operator from Point Clouds in R^d, 2009.
> > [4] Sharp et al. DiffusionNet: Discretization Agnostic Learning on Surfaces, 2020.
> > [5] Yadokoro et al. Weighted Combinatorial Laplacian and its Application to Coverage Repair in Sensor Networks, 2023.
> > [6] Lim, L.-H. Hodge Laplacians on Graphs, 2020.
> > [7] Nakahara, M. Geometry, Topology and Physics, 2003.

---

> > ### Author Rebuttal · Reviewer_BsYS · 2026-04-02
> >
> > Thank you for the detailed responses!
> >
> > As I personally do not have much background in the Hodge theorem, I will retain my current score and confidence interval!
> >
> > Thanks for introducing the theorem, and I will learn it accordingly.

---

> > > ### Author Response · Authors · 2026-04-02
> > >
> > > Thank you for confirming that our rebuttal has fully addressed your concern. In the paper, we include the theorem to provide theoretical completeness and to clarify the motivation behind the method, while keeping its practical value clear. We appreciate your consideration!

---

### Decision · Program_Chairs · 2026-04-30

**Decision:**

Accept (regular)

**Comment:**

The paper proposes a principled neural operator framework based on Hodge decomposition to preserve topological structure in PDE learning on manifolds.

Reviewers view the core idea as novel, mathematically well grounded, and empirically very strong, with particularly favorable results across multiple tasks and baselines.

While some concerns were raised about accessibility, notation, and aspects of the formalism, the rebuttal addressed these points well and strengthened the presentation and experimental positioning.